# Bioengineered amyloid peptide for rapid screening of inhibitors against main protease of SARS-CoV-2

Dongtak Lee [1,2,3,11], Hyo Gi Jung[1,4,11], Dongsung Park [1,5,11], Junho Bang[1,4], Da Yeon Cheong[6,7], Jae Won Jang[1,4], Yonghwan Kim[1,4], Seungmin Lee[1,8], Sang Won Lee[1,9], Gyudo Lee[6,7], Yeon Ho Kim[1,4], Ji Hye Hong[1,8], Kyo Seon Hwang [5] ✉, Jeong Hoon Lee [8] ✉ & Dae Sung Yoon [1,4,10] ✉

The coronavirus disease 2019 (COVID-19) caused by severe acute respiratory syndrome coronavirus-2 (SARS-CoV-2) has evoked a worldwide pandemic. As the emergence of variants has hampered the neutralization capacity of currently available vaccines, developing effective antiviral therapeutics against SARS-CoV-2 and its variants becomes a significant challenge. The main protease ($M^{pro}$) of SARS-CoV-2 has received increased attention as an attractive pharmaceutical target because of its pivotal role in viral replication and proliferation. Here, we generated a de novo $M^{pro}$-inhibitor screening platform to evaluate the efficacies of $M^{pro}$ inhibitors based on $M^{pro}$ cleavage site-embedded amyloid peptide (MCAP)-coated gold nanoparticles (MCAP-AuNPs). We fabricated MCAPs comprising an amyloid-forming sequence and $M^{pro}$-cleavage sequence, mimicking in vivo viral replication process mediated by $M^{pro}$. By measuring the proteolytic activity of $M^{pro}$ and the inhibitory efficacies of various drugs, we confirmed that the MCAP-AuNP-based platform was suitable for rapid screening potential of $M^{pro}$ inhibitors. These results demonstrated that our MCAP-AuNP-based platform has great potential for discovering $M^{pro}$ inhibitors and may accelerate the development of therapeutics against COVID-19.

The emergence of severe acute respiratory syndrome coronavirus-2 (SARS-CoV-2) has caused a global pandemic. The virus has exhibited a high rate of infectivity and the ability to spread rapidly across all age groups[1]. According to the World Health Organization (WHO), as of November 2023, there have been over 771 million confirmed infections and over 6.9 million cumulative deaths caused by the SARS-CoV-2[2].

Although vaccines against SARS-CoV-2 are being deployed worldwide, coronavirus disease 2019 (COVID-19) cases have increased because of the emergence of SARS-CoV-2 variants, such as the Delta and Omicron variants, which can break through the protective barrier of vaccination[3,4]. Various therapeutics have been developed targeting the spike proteins or viral RNA to treat COVID-19[5,6]. However, these drugs

[1]School of Biomedical Engineering, Korea University, Seoul 02841, South Korea. [2]Center for Nanomedicine, Department of Anesthesiology, Perioperative and Pain Medicine, Brigham and Women's Hospital, Boston, MA 02115, USA. [3]Harvard Medical School, Boston, MA 02115, USA. [4]Interdisciplinary Program in Precision Public Health, Korea University, Seoul 02841, South Korea. [5]Department of Clinical Pharmacology and Therapeutics, College of Medicine, Kyung Hee University, Seoul 02447, South Korea. [6]Department of Biotechnology and Bioinformatics, Korea University, Sejong 30019, South Korea. [7]Interdisciplinary Graduate Program for Artificial Intelligence Smart Convergence Technology, Korea University, Sejong 30019, South Korea. [8]Department of Electrical Engineering, Kwangwoon University, Seoul 01897, South Korea. [9]Terasaki Institute for Biomedical Innovation, Los Angeles, CA 90064, USA. [10]Astrion Inc, Seoul 02841, South Korea. [11]These authors contributed equally: Dongtak Lee, Hyo Gi Jung, Dongsung Park. ✉e-mail: k.hwang@khu.ac.kr; jhlee@kw.ac.kr; dsyoon@korea.ac.kr

have some limitations, such as low efficacy[7], various side effects due to frequent variations in the spiked protein[7], and mitochondrial RNA dysfunction in humans[8]. Therefore, there is an urgent need to identify and develop effective antiviral drugs against SARS-CoV-2 to combat this fatal disease.

Viral proteolytic enzymes, including human immunodeficiency virus-1 (HIV-1) and hepatitis C virus (HCV) NS3/4A proteases, play essential roles in viral proliferation and assembly, thereby making them promising potential therapeutic targets[9,10]. Among these, the main protease ($M^{pro}$), a cysteine protease containing a $His^{41}$–$Cys^{145}$ catalytic dyad[11], is an indispensable enzyme for SARS-CoV-2 replication and proliferation. The $M^{pro}$-mediated proteolytic post-processing of the SARS-CoV-2 replicase polyprotein, which cleaves at least 11 conserved sites, is crucial for viral assembly and maturation[11]. For example, $M^{pro}$ generates amyloidogenic proteins (e.g., spike protein and non-structural protein 11) in SARS-CoV-2 proteosomes with multiple aggregation-prone regions, facilitating viral self-assembly[12]. Considering the proven success of protease inhibitors in treating HIV-1 and HCV infections, strategies that specifically target $M^{pro}$ possess significant potential to thwart viral proliferation[10]. Importantly, as $M^{pro}$ has no known homologs in the human proteome, it is feasible to develop effective and selective $M^{pro}$ inhibitors without eliciting severe side effects[11]. Furthermore, the genetic similarity between the $M^{pro}$ of SARS-CoV-2 and SARS coronavirus (SARS-CoV, discovered in 2002) is 90%, whereas their genomes overall have only 79% similarity[1]. Given its critical role in the viral life cycle and the absence of closely related homologs in the human proteome, $M^{pro}$ is one of the most attractive targets for antiviral therapeutics to combat COVID-19. For instance, Paxlovid, an oral therapeutic that combines an $M^{pro}$ inhibitor (PF-07321332) with ritonavir, has garnered FDA approval for the treatment of moderate to severe COVID-19 cases[13]. Moreover, compounds like Pfizer's PF-07304814 and Simcere's SIM0417, designed to inhibit $M^{pro}$, are currently undergoing clinical trials as potential oral treatments against SARS-CoV-2[14–16].

Recently, drug screening methods have been developed to discover $M^{pro}$ inhibitors, such as virtual screening assays, fluorescence resonance energy transfer (FRET) assays, and cell-based assays[17,18]. However, screening for $M^{pro}$ inhibitors has been hampered by a high screening cost[19], poor reproducibility[20], and long screening cycles[20]. Although structure-based virtual screenings have been highlighted for repurposing and designing pharmaceutical drugs targeting $M^{pro}$, the practical use of these in silico methods is hindered by high false-positives due to the conformational flexibility of $M^{pro}$ and its protein-ligand dynamics[21]. Therefore, the development of highly rapid, cost-effective, and accurate drug screening methods is necessary to effectively identify medications against COVID-19.

Amyloids are misfolded proteins with a stable, unbranched, fibrous quaternary structure composed of repeating units of β-strands from protein or peptide monomers, and prone to easily self-assemble, forming highly organized fibrillar structure by intermolecular backbone hydrogen bonding[22]. Interestingly, owing to the well-organized supramolecular structure and exceptional physicochemical traits of amyloid proteins, they have been exploited in synthesizing functional hybrid nanocomposites containing gold nanoparticles (AuNPs), Quantum Dot, and graphene[23,24]. In particular, there has been some progress in developing protease-responsive nanomaterials that capitalize on the degradation of amyloid aggregates or fibrils. For example, Lee et al. synthesized amyloid corona by combining amyloid proteins and AuNPs as a drug screening platform for amyloid-β oligomer-degrading drugs[25]. In addition, Li et al. also reported the fabrication of highly conductive, biodegradable nanocomposites for biosensing applications to quantify enzymatic activity using complex structures composed of alternating layers of graphene and amyloid[26]. Therefore, considering the self-assembly properties of amyloid proteins, we speculated that a hybrid nanocomposite for drug screening can be synthesized with a combination of AuNPs and an amyloid peptide containing both $M^{pro}$ cleavage and amyloid-forming sequences. By leveraging both the catalytic activity and the localized surface plasmon resonance (LSPR) property of AuNPs, this hybrid nanocomposite is applicable to a high-throughput screening platform for $M^{pro}$ inhibitor candidates[23,25,27].

Here, we designed a bioengineered amyloid peptide containing both amyloid-forming sequences from prion protein (GNNQQY)[28] and the $M^{pro}$ cleavage site (LQS)[29] to develop a drug screening platform for discovering $M^{pro}$ inhibitors (Supplementary Fig. S1). We named our amyloid protein $M^{pro}$ cleavage site-embedded amyloid peptide (MCAP). We characterized the amyloid properties of MCAP using a thioflavin T (ThT) assay and atomic force microscopy (AFM). By combining MCAPs and AuNPs, we fabricated protease-sensitive nanocomposites, termed MCAP-AuNPs, wherein AuNPs were coated with the MCAP amyloid corona. The amyloid coronas on the surface of MCAP-AuNPs were easily degraded by $M^{pro}$, inducing the destabilization and aggregation of AuNPs. The aggregation of AuNPs caused a color change in an aqueous solution due to the intrinsic LSPR of AuNPs. This colorimetric change was used to evaluate the proteolytic activity of $M^{pro}$. By measuring the colorimetric response of MCAP-AuNPs, we evaluated the proteolytic activity of $M^{pro}$ and monitored the efficacy of various $M^{pro}$ inhibitors. Based on these investigations, we suggest that the MCAP-AuNP-based colorimetric screening system can be used to efficiently discover $M^{pro}$ inhibitors for combating COVID-19 and the infection of related coronaviruses (COVID-X) in the future.

## Results
### Fabrication and engineering of the $M^{pro}$ cleavage site-embedded amyloid peptide (MCAP)
Figure 1a shows the endosomal entry and replication process of SARS-CoV-2[30]. This process is initiated by binding between human angiotensin-converting enzyme-2 (hACE-2) receptor and spike protein (S protein) of SARS-CoV-2. Following these interactions, the S protein undergoes proteolytic cleavage by cathepsin L, leading to the conversion of the protein into a metastable state, subsequently triggering the fusion of the host cell membrane with the virus[31]. Subsequently, the genomic RNA of the virus is released into the cytoplasm, initiating the translation of co-terminal polyproteins (pp1a/ab). These polyproteins are then cleaved into non-structural proteins (nsps) by $M^{pro}$. The nsps product interacts with nsp12 to assemble the replicase-transcriptase complex, which is required for viral genome replication and the transcription of sub-genomic RNAs. Thus, $M^{pro}$ is essential for the survival of SARS-CoV-2.

Viral infection and replication in vivo involve polyproteins that are cleaved by $M^{pro}$ at L-Q ↓ (S, A, G) sequences, where ↓ marks the cleavage site[29]. Inspired by the interaction between $M^{pro}$ and polyproteins, we designed the MCAP (Fig. 1b). Specifically, $M^{pro}$ cleavage sequence (LQS) is inserted into the sequence of amyloid-forming peptides (GNNQQY). We speculate that MCAP may have amyloid-like properties and could be cleaved by $M^{pro}$, enabling us to analyze the proteolytic activity of $M^{pro}$ and the inhibitory effects of various drug candidates.

To confirm the amyloid-like properties of MCAP, the solution with MCAP monomers was incubated at 37 °C for 120 h. The results showed that MCAP fibrils were synthesized within five days, confirming the fibril-forming ability of MCAP (Supplementary Fig. S2). We analyzed the persistence length of each MCAP fibril using transmission electron microscopy (TEM) image and Image J software. The analysis showed that the persistence length of MCAP fibrils was $229.95 \pm 94.22$ nm, shorter than that of the original amyloid sequence (GNNQQY, $1.9 \pm 1.3\,\mu$m)[28]. In addition, the height of MCAP fibrils ($2.64 \pm 0.74$ nm, Supplementary Fig. S3) considerably decreased compared to that of the fibrils with original amyloid sequence ($43 \pm 24$ nm)[28]. These results support that the $M^{pro}$ cleavage sequences counterbalance the amyloid properties of MCAP.

 

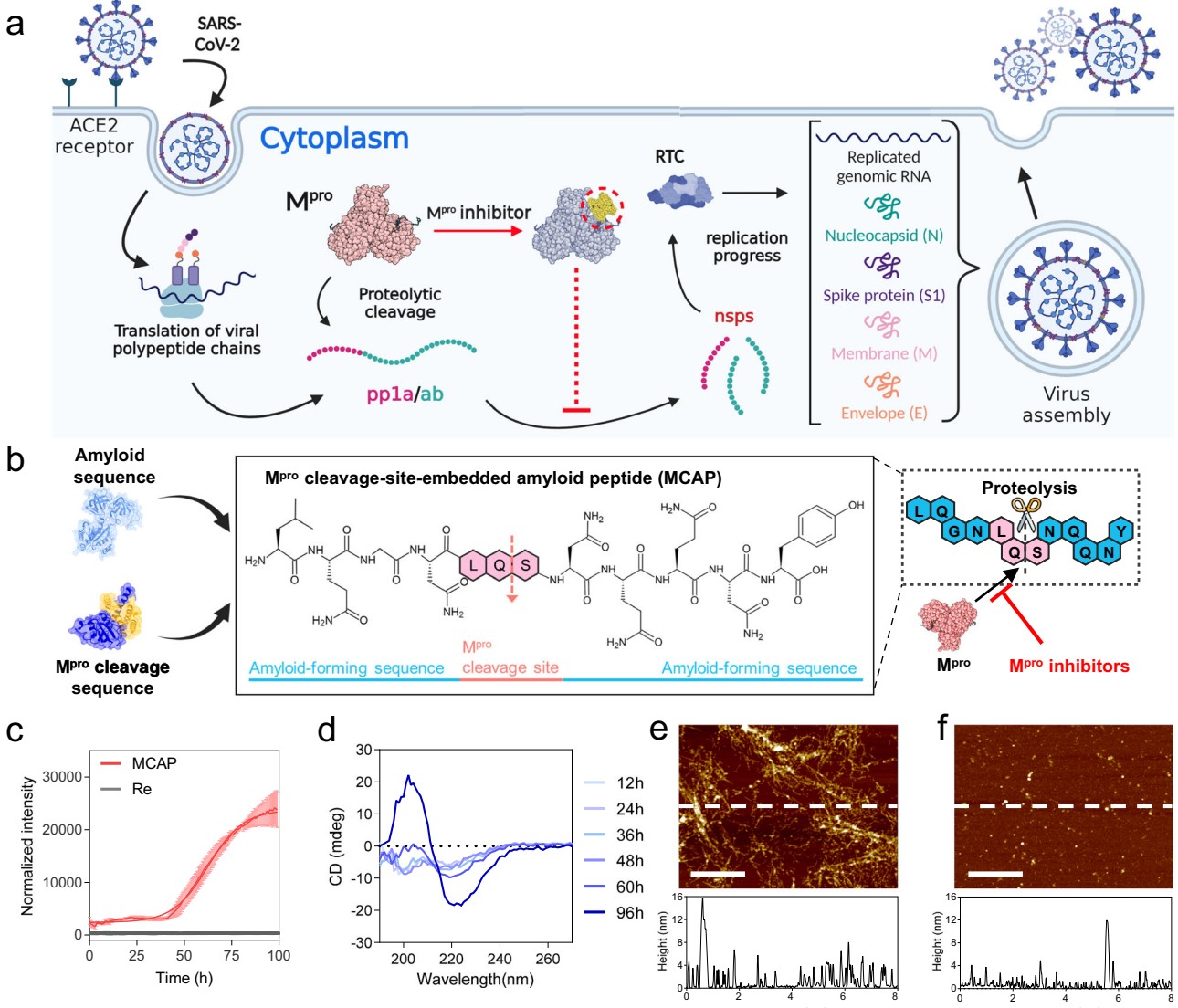

**Fig. 1 | Design of the main protease (Mpro) cleavage site-embedded amyloid peptide (MCAP) and evaluation of the amyloid characteristics of MCAP.**
**a** Schematic illustration of the replication of SARS-CoV-2 at the cellular level. Mpro inhibitors have the potential to prevent viral replication. The schematic illustration was created using BioRender.com. **b** Schematic illustration of MCAP comprising an amyloid sequence and an Mpro cleavage sequence. **c** ThT fluorescence measurement of MCAP and repeated Mpro cleavage sequence (Re, LQS-LQA-LQG-LQSS) with varying incubation time. The solid lines and error bars indicate the mean values and standard deviations derived from three independent experiments ($n = 3$). **d** Circular dichroism spectra of the MCAP solution upon incubation for over 96 h. **e** Height analysis of MCAP fibril using AFM. The bottom graph represents the cross-section height of the section indicated with the white dashed line. Scale bar: 2 μm. **f** Height analysis of MCAP fibrils incubated with Mpro for 12 h. Scale bar: 2 μm. The bottom graph represents the cross-section height of the section indicated with the white dashed line. The images of AFM in (**e**) and (**f**) were measured at least three times.

To investigate if the MCAP fibrils had a β-sheet structure, we first incubated MCAP monomers and monitored the kinetics of fibril formation by recording the fluorescence signal of ThT as they forms the β-sheet structure of amyloid fibrils (Fig. 1c). We observed a lag time of approximately 40 h, after which the growth phase was observed on the sigmoidal form of the ThT curve, strongly supporting the β-sheet formation of MCAP fibrils. In contrast, the peptide with repeated Mpro cleavage sequences (LQS-LQA-LQG-LQSS) only did not show any fluorescence signal for ThT compared to MCAP. Next, we performed circular dichroism (CD) analysis to monitor the β-sheet formation properties of MCAP depending on the incubation time (Fig. 1d). The results showed that the β-sheet peak gradually increased over time, validating the successful synthesis of MCAP fibrils.

Next, we incubated MCAP fibrils in the absence and presence of Mpro (0.1 mg mL⁻¹) and performed AFM analysis to determine the proteolytic ability of Mpro against MCAP fibrils. Without Mpro, the bundles

of MCAP protofibrils remain intact (Fig. 1e). In contrast, we observed that Mpro degraded MCAP fibrils into short fragments and small aggregates (Fig. 1f), confirming that Mpro successfully recognizes the Mpro cleavage sites in MCAP. The Fourier transform infrared (FT-IR) and CD analysis also revealed considerable reductions in β-sheet peaks of MCAP fibrils in the presence of Mpro, providing further evidence for the enzymatic degradation of MCAP fibrils by Mpro (Supplementary Fig. S4). Taken together, these results suggest that the bioengineered MCAP we generated possesses both amyloidogenic properties and enzymatic degradability due to the proteolytic activity of Mpro.

Furthermore, we designed engineered amyloid sequences MCAP2 (LQGN**LQA**NQQNY) and MCAP3 (LQGN**LQG**NQQNY) to explore the availability of different Mpro cleavage sequences (Supplementary Table S1). The AFM, TEM, and CD analysis confirmed the amyloidogenicity of MCAP2, ensuring β-sheet-rich fibril structures (Supplementary Fig. S5), while the MCAP3 showed no such structure

(Supplementary Fig. S6). Subsequently, the ThT analysis was conducted to confirm the degree of proteolytic activity of M[pro] at the MCAP, the MCAP2, and the MCAP3, respectively (Supplementary Figs. S10a–c). The analysis showed a 42% reduction in the ThT fluorescence intensity for the MCAP fibrils and a 22% reduction for the MCAP2 fibrils by M[pro]. In contrast, the MCAP3 aggregates exhibited no significant signal change after reacting with M[pro]. These results suggest that both variants of GNNQQY combined with LQS and LQA exhibit amyloidogenicity, and in particular, the MCAP containing LQS has the highest degradability to M[pro].

Interestingly, other types of amyloid peptides can also be used as basic amyloid-forming sequences instead of prion sequences. To explore the availability of other amyloid-forming sequences, we designed various engineered sequences derived from amyloid-forming sequences such as islet amyloid polypeptide (IAPP, SNNFGAIL)[32] and amyloid-β (KLVFFAE, GGVVIA)[33]. The IAPP-derived engineered amyloid sequence is SNLQSNFGAIL (i.e., IAPP MCAP), and the amyloid-β-derived sequences are KLLQSVFFAE (i.e., Aβ MCAP1) and GGLQSVVIA (i.e., Aβ MCAP2) (Supplementary Table. S1). To confirm the amyloidogenicity of each engineered peptide, we conducted AFM, TEM, and CD analysis (Supplementary Figs. S7–S9). The analyses showed that each engineered sequence formed fibrous aggregates with β-sheet structures. To assess the reactivity between M[pro] and fibrils synthesized with each amyloid sequence, a ThT assay was conducted (Supplementary Fig. S10). The results showed that after reacting with M[pro], the ThT fluorescence intensity of the MCAP, IAPP MCAP, Aβ MCAP1, and Aβ MCAP2 fibrils decreased by 42%, 26%, 17%, and 23%, respectively. These results indicate that the MCAP possesses superior sensitivity in assessing the proteolytic activity of M[pro] compared to other engineered amyloid sequences.

To investigate whether the core sequence of MCAP is cleaved by M[pro] in fibrils, we strategically repositioned the M[pro]-cleavage sequence and designed an engineered sequence (Repositioned MCAP, LQGNNQQNYLQG) (Supplementary Table S1). The AFM, TEM, and CD analysis revealed that the Repositioned MCAP formed fibrous aggregates with β-sheet structure (Supplementary Fig. S11). The reactivity test between M[pro] and Repositioned MCAP fibrils, assessed through AFM analysis, revealed that the fibrillar structure remained unaffected by M[pro] treatment (Supplementary Fig. S12). In addition, the CD spectra of Repositioned MCAP fibrils treated with M[pro] exhibited negligible peak change (~10% reduction in ThT fluorescence) of the β-sheet structures, compared to the untreated fibrils. Based on these results, we confirmed that M[pro] actively degrades only the MCAP fibrils that contain the M[pro]-cleavage sequence at the core region of the engineered sequence.

## Fabrication of MCAP amyloid corona using AuNPs

To synthesize MCAP amyloid corona with uniform size and morphology, we used AuNPs as the nucleation site of MCAP and fabricated MCAP-AuNP comprising MCAP amyloid corona (Fig. 2a). The negatively charged surface of AuNPs attracted the positively charged N-terminal of MCAP, leading to an increasing local concentration of MCAPs on the AuNP surface[23]. The increased concentration promotes the events of the formation of β-sheet structure and facilitates the nucleation of MCAPs on AuNPs. After 24 h incubation, amyloid corona was synthesized on AuNPs (MCAP-AuNP). Amyloid coronas on AuNPs predominantly tend to be hard protein coronas because they are interwoven with each other and irreversibly bind to the surface of AuNPs[25,34]. The amyloid corona can provide several properties to AuNPs, including steric stabilization[35,36], antifouling[37], and high salt resistance[23,25].

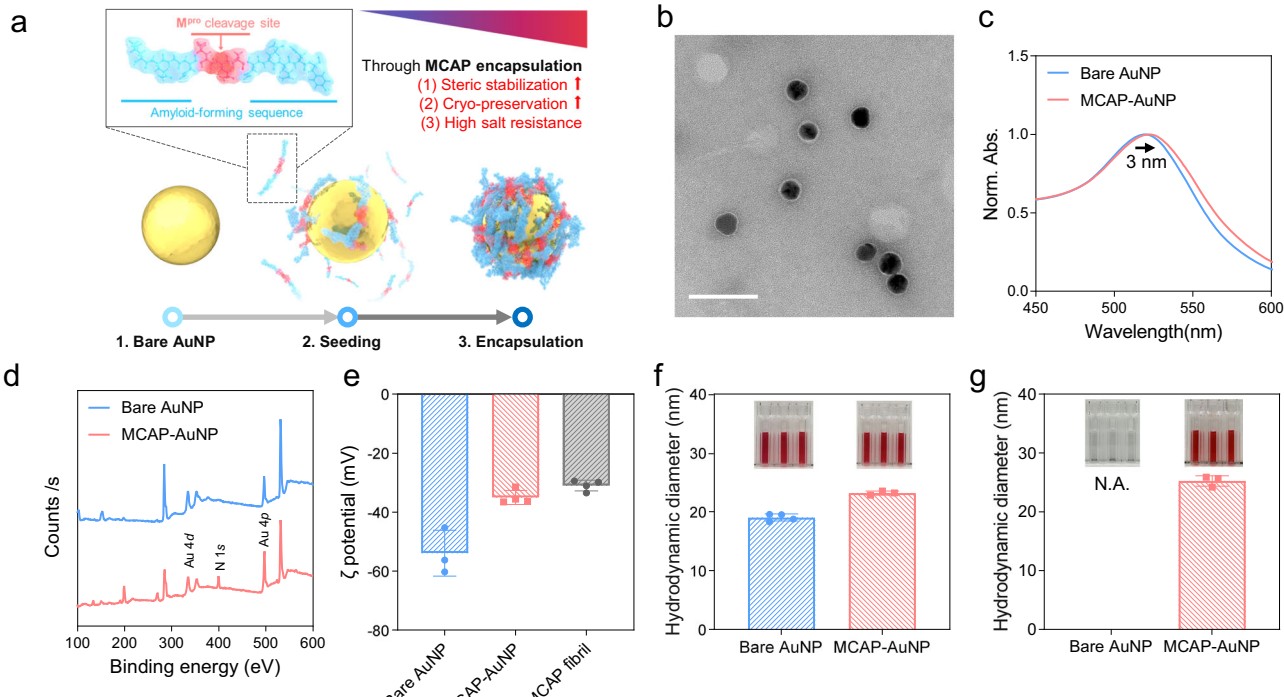

**Fig. 2 | Physicochemical characteristics of MCAP-AuNPs. a** Schematic illustration of the fabrication process of MCAP-AuNPs. **b** TEM image of MCAP-AuNPs. The white scale bar is 50 nm in size. TEM images were measured at least three times. **c** UV-vis spectra of MCAP-AuNPs and bare AuNPs. The spectra peak was shifted 3 nm from bare AuNPs to MCAP-AuNPs. **d** XPS analysis of MCAP-AuNPs and bare AuNPs. The N 1 s peak was only measured in MCAP-AuNPs. **e** ζ-potential of the bare AuNPs, the MCAP-AuNPs, and the MCAP fibrils Data are presented as mean values ± standard deviation (SD) from $n = 3:4:4$ independent experiments. The hydrodynamic diameter of bare AuNPs and MCAP-AuNPs (**f**) before and (**g**) after freezing. Data are presented as mean values ± SD from $n = 4$ independent experiments for bare AuNPs and $n = 3$ for MCAP-AuNPs, respectively. The inserted pictures are actual photographs of each sample.

We optimized the stability of the MCAP-AuNPs by varying the initial concentration of MCAP monomer (Supplementary Fig. S13). At a low concentration of MCAP monomer (<4.04 μM), the ultraviolet-visible (UV-vis) spectra of MCAP-AuNPs shifted from purple to red, because the MCAP monomers cannot fully cover the surface of AuNPs, inducing only a partial aggregation of AuNPs. Similarly, the extent of AuNP aggregation increased at higher concentrations of MCAP monomer (>4.04 μM). Thus, we determined that the optimal concentration of MCAP monomer was 4.04 μM because it was the concentration at which MCAP fully covered the surface of the AuNPs with a high yield (~98%, Supplementary Fig. S14). Moreover, good steric stability was observed in that setup for several weeks. Therefore, we confirmed that the MCAP amyloid corona was homogeneously coated on the surface of each AuNP, revealing long-term steric stability (Fig. 2b and Supplementary Fig. S14). In addition, the FT-IR analysis of the MCAP-AuNPs showed that the MCAP aggregates on the surface of AuNPs have β-sheet-rich conformation (Supplementary Fig. S15).

## Physicochemical characteristics of the fabricated MCAP-AuNPs

The LSPR peak in the UV spectra of bare AuNPs and MCAP-AuNPs shifted from ~520 nm to ~523 nm (Fig. 2c). This result indicates that the MCAP amyloid corona covered the surface of the AuNPs, partially increasing the local refractive index of the MCAP-AuNPs[38]. X-ray photoelectron spectroscopy (XPS) also confirmed the encapsulation of the MCAP amyloid corona on AuNPs because, when compared to the spectrum of bare AuNPs, only the MCAP-AuNP spectrum showed N 1 $s$ peaks (~400 eV) (Fig. 2d). The zeta potential of the bare AuNPs, MCAP-AuNPs, and MCAP fibrils were −53.92 mV, −36.54 mV, and −30.96 mV, respectively. The results indicate that the MCAP-AuNPs are more similar in zeta potential with the MCAP fibrils as compared to the bare

AuNPs, thereby supporting successful MCAP-coating on the surface of AuNPs (Fig. 2e).

To verify the stability of the MCAP-AuNPs, we conducted a temperature-dependent stability test and a freeze-thaw test of the MCAP-AuNPs. The temperature-dependent stability test showed that the MCAP-AuNPs were stable at 40 °C but unsteady at 60 °C, thereby inducing particle aggregation due to thermal denaturation (Supplementary Fig. S16). Before freezing, the hydrodynamic diameters of the bare AuNPs and MCAP-AuNPs were 19.02 ± 0.53 nm and 23.18 ± 0.29 nm, respectively, which were consistent with the result in Fig. 2c and Supplementary Fig. S14 (Fig. 2f). The MCAP-AuNPs were ~4 nm larger than the bare AuNPs, corresponding to the TEM image in Fig. 2b. After freeze-thawing, the bare AuNPs aggregated, and their hydrodynamic diameters were not observed (Fig. 2g). In contrast, the MCAP-AuNPs retained their initial hydrodynamic diameters after the freeze-thaw process and were durable up to the fourth cycle of the freeze-thaw test (Supplementary Fig. S17), indicating that the MCAPs were irreversibly bound to AuNP surface as a hard corona[25,39]. These results also suggest that the MCAP-AuNPs have good cryopreservation storage capacity that prevents the denaturation of MCAPs.

## Proteolytic activity of M^pro as measured via MCAP-AuNPs

Figure 3a shows our MCAP-AuNP-based strategy for monitoring the proteolytic activity of M^pro. When M^pro is added to an MCAP-AuNP solution, M^pro degrades the MCAPs on the AuNP surface, exposing the surface of bare AuNPs and promoting their aggregation under physiological conditions[40]. The aggregation of AuNPs leads to an LSPR shift in its spectrum, causing a color change from red to purple. However, inactivated M^pro cannot degrade the MCAP on the surface of

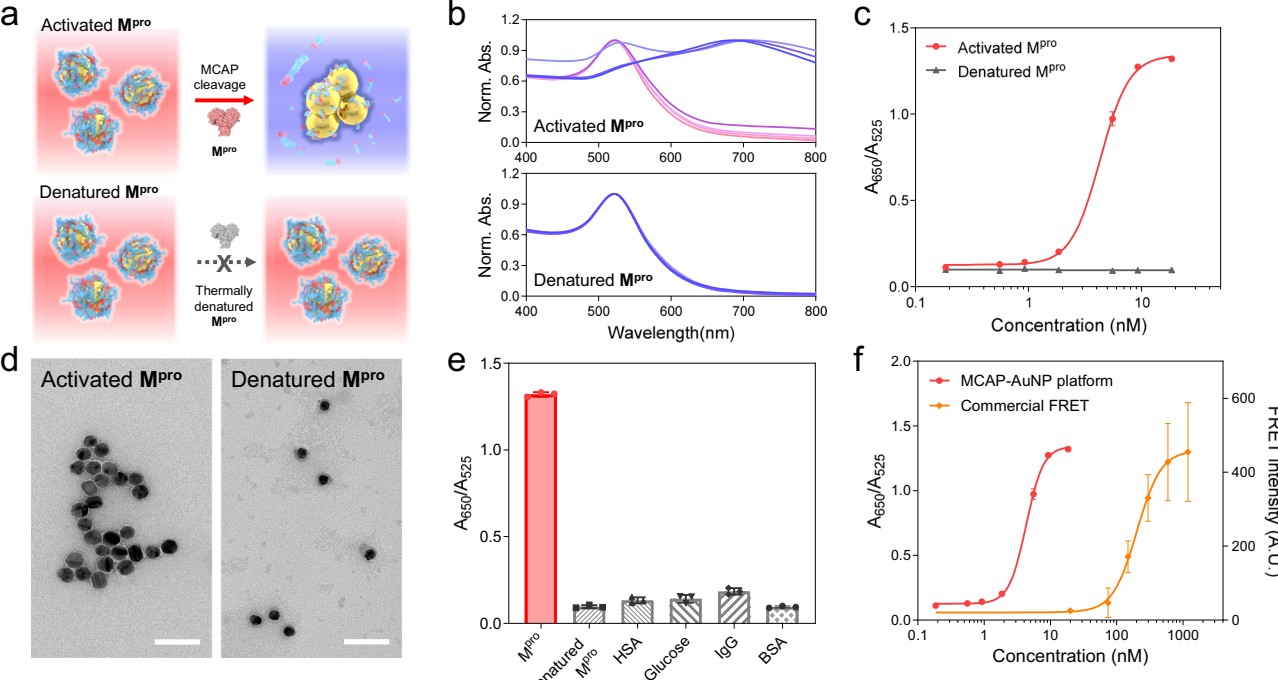

**Fig. 3 | Analysis of the proteolytic activity of M^pro using MCAP-AuNPs.**
**a** Schematic illustration of the principle of measuring the proteolytic activity of M^pro using MCAP-AuNPs. M^pro degrades the MCAP molecules on the AuNP, inducing particle aggregation. However, denatured M^pro cannot induce particle aggregation because it does not have proteolytic activity. **b** UV–vis spectra of MCAP-AuNPs reacting with various concentrations (1–7: 0.19, 0.56, 0.93, 1.85, 5.55, 9.25, and 18.5 nM) of intact M^pro (upper) and denatured M^pro (bottom). **c** Plots fitted using the sigmoidal dose-response curve as a function of M^pro concentration. **d** TEM image of

MCAP-AuNPs that reacted with M^pro (left) and denatured M^pro (right). The size of white scale bar is 50 nm. **e** Selectivity test of MCAP-AuNP. The relative ratio ($A_{650}/A_{525}$) of MACP-AuNPs was measured after reacting with M^pro, denatured M^pro, HSA, glucose, IgG, and BSA for 24 h. **f** Concentration-dependent M^pro activity test using MCAP-AuNP platform and FRET-based assay. All data in (**c**), (**e**), and (**f**) are presented as mean values ± SD from $n = 3$ independent experiments. Error bars are not shown where they are smaller than the circle symbols.

AuNPs. Thus, MCAP-AuNPs remain intact in the solution, whose color does not change. Based on this strategy, we quantitatively measured the proteolytic activity of M$^{pro}$.

To validate our strategy by measuring the proteolytic activity of M$^{pro}$, we quantified the degree of MCAP degradation by intact and denatured M$^{pro}$ using UV-vis spectra (Fig. 3b). The results revealed that the MCAP-AuNP solution exhibited an LSPR shift from red to purple depending on the M$^{pro}$ concentration (0.19, 0.56, 0.93, 1.85, 5.55, 9.25, or 18.5 nM). A higher concentration of M$^{pro}$ induced a larger red shift in the LSPR, indicating that the MCAP-AuNPs aggregated actively at high M$^{pro}$ concentrations. In contrast, for denatured M$^{pro}$, which was incubated at 90 °C for 4 h, the UV-vis spectra of the MCAP-AuNPs remained unchanged (Fig. 3b). This indicates that denatured M$^{pro}$ cannot degrade the MCAP-AuNPs, resulting in no particle aggregation of them.

For the UV-vis spectra of the MCAP-AuNP solution, the absorbances at $A_{650}$ and $A_{525}$ represent the degrees of MCAP-AuNP aggregation and dispersion, respectively. Thus, the relative absorbance ratio $A_{650}/A_{525}$ was adopted to quantify the degree of MCAP-AuNP aggregation. We found that the $A_{650}/A_{525}$ ratio of the MCAP-AuNP solution increased depending on the M$^{pro}$ concentration (Fig. 3c). In contrast, a concentration-dependent $A_{650}/A_{525}$ ratio change was not observed using denatured M$^{pro}$. These results correspond with the TEM images in Fig. 3d. The TEM images showed that an increased aggregation of MCAP-AuNPs took place at high concentrations of M$^{pro}$ and no aggregation at denatured M$^{pro}$. To quantitatively represent the proteolytic activity of M$^{pro}$, we analyzed the $A_{650}/A_{525}$ ratio of MCAP-AuNP solution as a function of M$^{pro}$ concentration using a sigmoidal dose-response model, shown in Eq. (1) below.

$$A_{650}/A_{525} = 0.127 + \frac{1.215}{1 + 10^{Log 4.297 - [M^{pro}]}}, R^2 = 0.99 \tag{1}$$

From this equation, the half-maximal effective concentration (EC$_{50}$) was 4.297 nM and the maximal efficacy was 1.215 (a.u.). This suggests that 4.297 nM of M$^{pro}$ is sufficient to mediate the massive aggregation of MCAP-AuNPs. We also investigated the concentration- and time-dependent proteolytic activity of M$^{pro}$ induced by the MCAP-AuNPs (Supplementary Fig. S18). A red shift in the UV spectra was observed at higher M$^{pro}$ concentrations and reaction times. The selectivity tests of MCAP-AuNPs were conducted with various interfering biomolecules abundant in physiological conditions, such as bovine serum albumin (BSA), immunoglobulin, glucose, and human serum albumin (HSA). The results showed that the MCAP-AuNPs were not affected by these biomolecules and selectively reacted with M$^{pro}$ (Fig. 3e and Supplementary Fig. S19). Thus, we concluded that MCAP-AuNPs are a suitable platform for measuring the proteolytic activity of M$^{pro}$.

We also measured M$^{pro}$ activity using the commercialized FRET substrate MCA-AVLQSGFR-Lys(Dnp)-Lys-NH$_2$ trifluoroacetate (Sigma Aldrich, USA) (Fig. 3f). We obtained the EC$_{50}$ values of the FRET system from the sigmoidal dose-response curve as a function of M$^{pro}$ concentration. The EC$_{50}$ of the FRET system was estimated to be 211 nM, which is approximately 50 times higher compared to that of our platform (4.4 nM). Furthermore, we observed that the MCAP-AuNP-based system displayed a much lower variance for each data point, compared to the FRET-based system. These results suggest that our MCAP-AuNP-based system offers superior sensitivity and accuracy in measuring M$^{pro}$ activity as compared to the FRET-based system. Additionally, it requires fewer amounts of expensive enzymes, such as M$^{pro}$, for screening.

In our MCAP-AuNP platform, a very small portion of the MCAPs exists in the form of free MCAPs in the supernatant of the MCAP-AuNP solution. These free MCAPs may affect M$^{pro}$ activity. To scrutinize the effect of free MCAPs, we conducted a M$^{pro}$ activity test of the MCAP-AuNP solutions with and without free MCAPs (Supplementary Fig. S20). Firstly, we ensured that the free MCAPs were fully removed after centrifugation of the MCAP-AuNP solution. Then, the M$^{pro}$ activity test was conducted. The results showed that the as slightly shifted from 3.5 nM to 2.1 nM in the presence of the free MCAPs. This behavior indicates that the majority of MCAPs are irreversibly bound to the surface of AuNPs as hard corona. The existence of a small amount of free MCAPs slightly changes the apparent activity of M$^{pro}$. For more precise screening, it is recommended to eliminate the free MCAPs prior to drug screening.

## Molecular docking analysis of M$^{pro}$ and its potential inhibitors

Before screening potential inhibitors with MCAP-AuNPs, we conducted molecular docking to evaluate their interaction with the active sites of M$^{pro}$. The active site cleft between domains I and II houses the catalytic dyad His$^{41}$-Cys$^{145}$, which is known to play a critical role in the proteolytic activity of M$^{pro}$[41]. Based on our analysis, we selected four competitive inhibitors with high binding affinities that directly bind to this crucial catalytic active site for further experiments. We compared their pharmacokinetic properties obtained through molecular docking simulations and the results from the MCAP-AuNPs-based screening platform.

Figure 4 shows the interacting residues and molecular structures of the binding pocket of M$^{pro}$ with ebselen, leupeptin, hesperetin, and lopinavir. Ebselen is known for its antioxidative, anti-inflammatory, and cytoprotective properties against COVID-19 and has also been investigated for other diseases, such as hearing loss and bipolar disorders[42]. Figure 4a shows the molecular docking of ebselen and M$^{pro}$. At the catalytic site of M$^{pro}$, the carbonyl oxygen of ebselen interacts with the Asn$^{142}$ and Gln$^{189}$ side chains of M$^{pro}$ through hydrogen bonding. In addition, hydrophobic contacts between ebselen and the His$^{41}$, Cys$^{145}$, Met$^{165}$, Pro$^{168}$, Met$^{49}$, and His$^{164}$ of M$^{pro}$ can be made[43]. The interaction between the catalytic dyad of M$^{pro}$ and ebselen inhibits the proteolytic activity of M$^{pro}$.

For leupeptin, a well-known covalent inhibitor of threonine, cysteine, and serine proteases[44], its C-terminal aldehyde group reacts with the Cys$^{145}$ of M$^{pro}$ to form a hemithioacetal[45] (Fig. 4b). In addition, leupeptin forms a water-mediated interaction with the side chain of Glu$^{189}$ and hydrogen bonds with the main chain of His$^{164}$ and Cys$^{145}$ in M$^{pro}$. These interactions can reduce the stability of the M$^{pro}$-leupeptin complex[45]. Similarly, hesperetin, which has neuroprotective effects against neurodegenerative diseases, forms a single hydrogen bond with the His$^{41}$, Leu$^{141}$, Cys$^{145}$, Glu$^{166}$, Arg$^{188}$, and Thr$^{190}$ of M$^{pro}$ (Fig. 4c)[46]. Lopinavir, which is used to treat HIV infection, interacts with the His$^{41}$, Cys$^{145}$, Gln$^{189}$, Met$^{164}$, Met$^{49}$, Glu$^{166}$, and Leu$^{27}$ residues of M$^{pro}$ (Fig. 4d). Thus, the interaction between the inhibitors and residues of M$^{pro}$, especially the active sites His$^{41}$ and Cys$^{145}$, has the possibility to reduce the proteolytic activity of M$^{pro}$.

Next, we evaluated the docking binding free energy ($\Delta G_{Dock}$) between four potential inhibitors and the active sites of M$^{pro}$ to characterize the protein-ligand association. The results revealed that hesperetin (−7.3 kcal/mol) has the highest binding affinity, followed by ebselen (−6.6 kcal/mol), lopinavir (−6.4 kcal/mol), and leupeptin (−6.0 kcal/mol). These values are consistent with those reported in previous studies[47–50]. Inhibition constants ($K_i^{MD}$), representing the potency of each inhibitor, were calculated as below;

$$K_i^{MD} = e^{\frac{\Delta G_{Dock}}{RT}} \tag{2}$$

where, $R$ is the gas constant ($1.987 \times 10^{-3}$ kcal K$^{-1}$mol$^{-1}$) and $T$ is the temperature in Kelvin (298.15 K). The $K_i^{MD}$ values of hesperetin, ebselen, lopinavir, and leupeptin were determined to be 4.45 μM, 14.51 μM, 20.34 μM, and 39.95 μM, respectively. The $\Delta G_{Dock}$ and $K_i^{MD}$ values obtained from the docking of selected potential inhibitors against M$^{pro}$ are summarized in Supplementary Table S2.

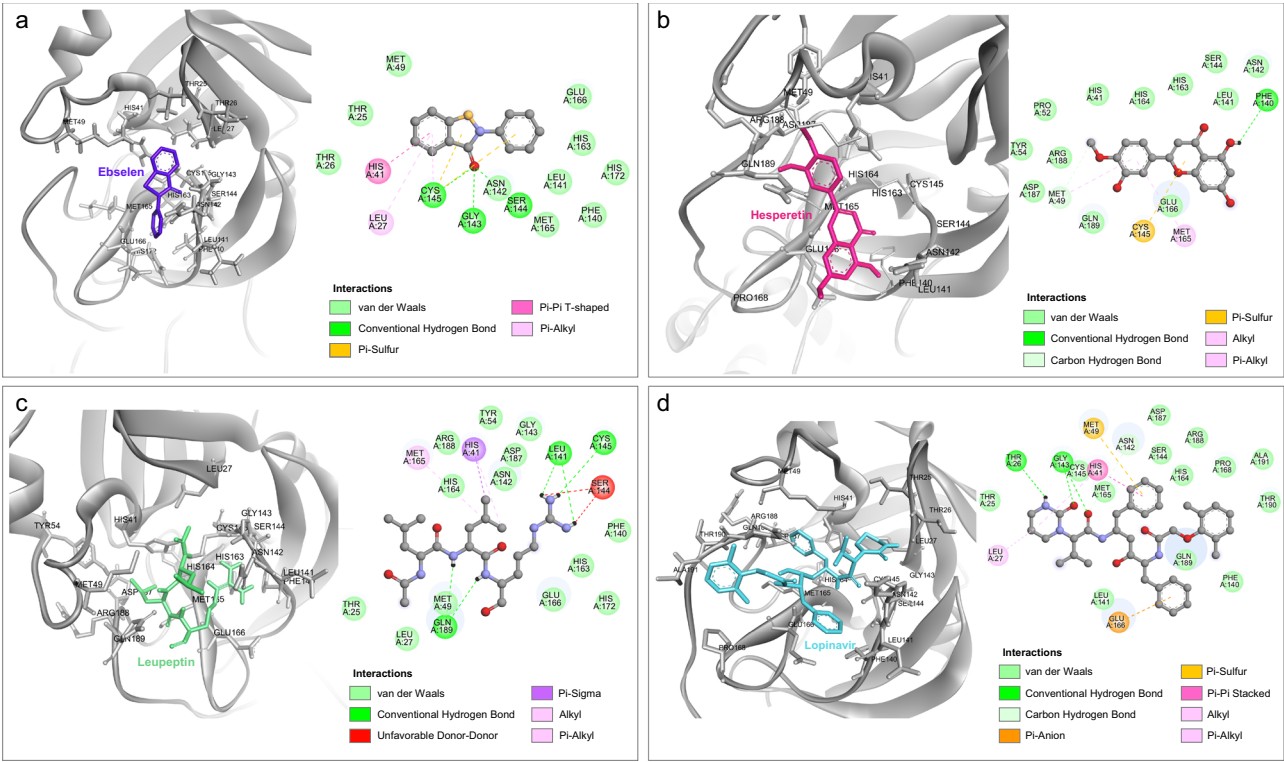

**Fig. 4 | Comparison of the structural basis and molecular mechanisms of M^pro inhibitors.** Molecular structure of the binding pockets of M^pro (PDB ID: 6LU7) with (**a**) ebselen, (**b**) leupeptin, (**c**) hesperetin, and (**d**) lopinavir. Orange, gray, blue, red, and black balls indicate selenium, carbon, nitrogen, oxygen, and hydrogen atoms, respectively. Hydrogen atoms bonded to carbon atoms are omitted for clarity.

## MCAP-AuNP-based platform for M^pro inhibitor screening

To validate the efficacy of M^pro inhibitors, we designed an MCAP-AuNP-based screening platform and measured the inhibition efficacy of ebselen, leupeptin, hesperetin, and lopinavir against M^pro using MCAP-AuNPs. Figure 5a shows the working principle of the MCAP-AuNP-based M^pro inhibitor screening platform. In detail, the M^pro-induced proteolysis of MCAP encapsulated on the AuNP surface caused the sterically stabilized MCAP-AuNPs to become unstable and prone to aggregation. Inhibitors that bind to the proteolytic active site of M^pro hamper the M^pro-induced degradation of MCAPs on the surface of AuNPs. The degree of MCAP-AuNP aggregation depended on the state of M^pro inactivation, which was proportional to the concentration of M^pro inhibitors. To investigate the efficacy of the inhibitors, we measured the degree of MCAP-AuNP aggregation using UV-vis spectroscopy (Fig. 5b). In our drug screening platform, the concentration of M^pro was kept constant (18.5 nM), and each M^pro solution was pre-incubated with various concentrations of inhibitors for 20 min before reacting them with MCAP-AuNPs.

The LSPR shifts of the MCAP-AuNP solutions decreased depending on the concentration of various inhibitors (Supplementary Fig. S21). Because each inhibitor has different efficacies, we defined the inhibition ratio (%) to quantify the inhibition efficacy of each inhibitor as follows:

$$Inhibition\ ratio\ (\%) = \frac{A_{525}/A_{650}}{Maximum\ (A_{525}/A_{650})} \quad (3)$$

where $A_{650}$ and $A_{525}$ are the absorption spectra at 650 and 525 nm, respectively, and the maximum $A_{525}/A_{650}$ is the absorbance when the MCAP-AuNPs are not aggregated.

We also analyzed the inhibition ratio as a function of inhibitor concentration using a sigmoidal dose-response model (Fig. 5c–f). The equations generated for each inhibitor are listed in Supplementary Table S3. From these equations, we determined the half-maximal inhibitory concentration (IC$_{50}$) values of ebselen, leupeptin, hesperetin, and lopinavir, which were 0.39 μM, 28.57 μM, 43.14 μM, and 10.7 μM, respectively (Fig. 5h). Among these four clinical trial drug candidates, ebselen exhibited the strongest M^pro inhibition activity (IC$_{50}$ = 0.39 μM), consistent with previous studies[11]. In addition, we also compared the inhibitory efficacy of two promising antiviral flavonoids, hesperetin and hesperidin, using our platform. Hesperidin, a glycoside containing rutinose (α-L-rhamno-pyranosyl-[1→6]-β-D-glucopyranose) linked to the OH-7 of hesperetin, exhibited an IC$_{50}$ value of 369.4 μM, which was much higher than that of hesperetin (43.14 μM) (Supplementary Fig. S22). This is consistent with previous studies showing that hesperetin has a more substantial inhibitory effect on M^pro than hesperidin[51]. From these results, we verified that our MCAP-AuNP-based screening platform has excellent capability for validating and quantifying the drug efficacy of M^pro inhibition.

The enzyme kinetics of M^pro was controlled by its inhibitors, so the reaction rate between M^pro and MCAP-AuNP was evaluated depending on the MCAP-AuNP concentration (Fig. 5g–j). This can be represented by Michaelis-Menten model as follows:

$$V_0 = V_{max} \times \frac{[S]}{K_M + [S]} \quad (4)$$

where $V_0$ is the initial velocity of the reaction, $V_{max}$ is the maximum velocity of the reaction, $K_M$ is the Michaelis constant, and $[S]$ is the MCAP-AuNP concentration.

Here, the $V_{max}$ and $K_M$ were determined depending on the inhibitor concentration (Fig. 5k). We found that the reaction rate tended to decrease at higher inhibitor concentrations. At a high concentration of each inhibitor (2.27 μM ebselen, 81.78 μM leupeptin, 13.6 μM hesperetin, and 27.26 μM lopinavir), the resulting $V_{max}$ was approximately

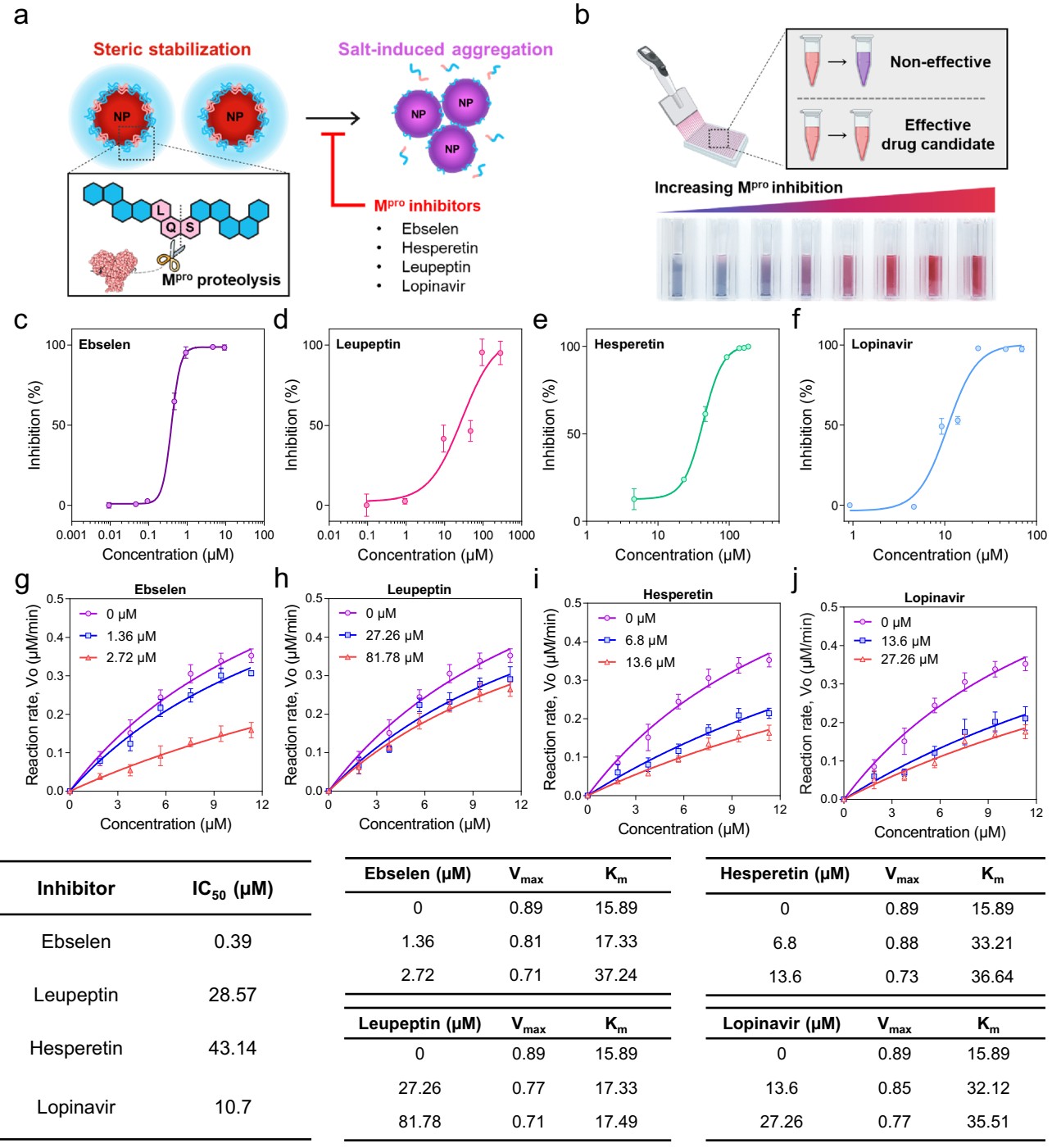

**Fig. 5 | Evaluation of the Mpro inhibition efficacy of various drugs using the MCAP-AuNP-based screening platform generated in this study. a** Schematic illustration of the principle of measuring the inhibition efficacy of drugs using MCAP-AuNP. The degree of MCAP-AuNP aggregation was measured as the inhibitor-controlled Mpro activity. **b** Schematic illustration of measuring the efficacy of Mpro inhibitors using the colorimetric change in MCAP-AuNP solutions. The colorimetric change of the solution represents the inhibition efficacy of the drug against Mpro. The bottom image shows the actual photos of MCAP-AuNP solutions mixed with various concentrations of hesperetin (0, 4.6, 23.1, 46.3, 92.5, 138.8, 161.9, and 185 μM). The dose-dependent inhibition ratio (%) curves of the MCAP-AuNP solution depending on various concentrations of Mpro inhibitors: (**c**) ebselen, (**d**) leupeptin, (**e**) hesperetin, and (**f**) lopinavir (*n* = 3 for each). Michaelis-Menten plots of MCAP-AuNP solutions treated with various Mpro inhibitors: (**g**) ebselen, (**h**) leupeptin, (**i**) hesperetin, and (**j**) lopinavir. All data are presented as mean values ± SD from *n* = 3 independent experiments. Error bars are not shown where they are smaller than the circle symbols. The bottom table contains the fitting values for each graph.

1.18 ~ 1.2-fold lower than that of non-inhibited Mpro. As for ebselen, a relatively low concentration (1.36 μM) decreased the reaction rate, consistent with its relatively low IC$_{50}$ value (0.39 μM). The TEM image also confirmed the inhibitory effects of each inhibitor in Supplementary Fig. S23. These results indicate that our MCAP-AuNP-based system can be used to evaluate the proteolytic activity of Mpro as controlled by its inhibitors.

We determined the values of experimental inhibition constant ($K_i^{Exp}$) for the four potential inhibitors utilizing the IC$_{50}$ and $K_M$ values obtained from the results with our screening platform. The four

inhibitors are competitive and directly interact with the active site of $M^{pro}$. Therefore, we calculated $K_i^{Exp}$ values using $IC_{50}$ and $K_M$ values in the following equation[52].

$$K_i^{Exp} = \frac{IC_{50}}{1 + \frac{[S]}{K_M}} \quad (5)$$

Where $IC_{50}$ is the half-maximal inhibitory concentration, $K_M$ is the Michaelis constant, and $[S]$ is the MCAP-AuNP concentration. The $K_i^{Exp}$ values of ebselen, leupeptin, hesperetin, and lopinavir were determined to be 0.23 µM, 16.69 µM, 25.19 µM, and 6.25 µM, respectively. In contrast to the results of molecular docking simulations, ebselen exhibited the lowest $K_i^{Exp}$ value, indicating the highest binding affinity to $M^{pro}$ in the experiment. The $K_i^{Exp}$ values, along with $K_i^{MD}$ values obtained from the docking simulation for selected potential inhibitors against $M^{pro}$, are summarized in Supplementary Table S2. The results revealed that $K_i^{Exp}$ values differed by about 0.17 to 63.71 times compared to each $K_i^{MD}$ value, demonstrating that the theoretically calculated binding affinity cannot perfectly reflect the experimental value[21]. Our MCAP-AuNP-based screening platform can overcome the limitations of molecular docking by providing precise experimental validation for pharmacokinetic analysis. Taken together, we confirmed that our platform provides a potential means for precisely evaluating both the $M^{pro}$ activity in the presence of an inhibitor and its drug efficacy in a fast and label-free manner.

## Discussion

Despite the rollout of various COVID-19 vaccines, accumulating evidence indicates that the efficacy of these vaccines has diminished against new SARS-CoV-2 variants (e.g., Delta and Omicron) owing to frequent mutations in the S protein[53]. Therefore, there is an urgent need to develop antiviral drugs to combat COVID-19 and complement the therapeutic effect of currently available preventive vaccines. $M^{pro}$, an essential enzyme for viral replication and proliferation, is an outstanding pharmaceutical target for SARS-CoV-2 because it is highly conserved among all coronaviruses[6].

Recent studies have revealed that small molecules that bind to the catalytic dyad (His[41] and Cys[145]) and inhibit the proteolytic activity of $M^{pro}$ are desirable as antiviral therapeutics. Various strategies have been employed to identify $M^{pro}$ inhibitors that can bind to its catalytic dyad, including in silico drug discovery tools such as molecular docking[54–56]. Despite the high performance of structure-based virtual screening techniques, these methods are limited by several inherent drawbacks. Specifically, the limited consideration of protein flexibility and conformational changes poses significant challenges in predicting the accurate pharmacokinetic properties of $M^{pro}$ inhibitors[21]. Therefore, experimental validation of the docking results is crucial for developing antiviral drugs with adequate pharmacokinetic analysis. Despite numerous papers on structure-based virtual screening of potential $M^{pro}$ inhibitors, only few studies have reported experimental confirmation to validate their docking results[11,57]. The lack of empirical evaluation is mainly due to the limited access to biological facilities and the lack of in vitro screening techniques. Moreover, SARS-CoV-2 infection and propagation in cell culture require qualified professionals and must be performed in biosafety level-3 (BSL-3) facilities[58]. Given these critical bottlenecks in the conventional methods, a novel strategy is necessary to develop a rapid, simple, and label-free drug screening platform.

In this study, we developed an MCAP-AuNP-based colorimetric screening platform to discover $M^{pro}$ inhibitors. Our platform allows rapid (<2 h), label-free, real-time monitoring of drug efficacy, simultaneously across hundreds of compounds with minimal enzyme usage (<10 nM). To optimize our high-throughput screening platform, we considered three crucial factors. First, it should be simple and robust during fabrication and assay processes. We designed the MCAP

sequence using the convergence of amyloidogenic and $M^{pro}$-cleavable sequences. We confirmed that the bioengineered MCAP sequence facilitates the formation of amyloid fibrils through self-assembly and that $M^{pro}$ can readily degrade these self-assembled MCAP sequences. Furthermore, our findings demonstrated that the MCAP exhibits superior performance compared to other engineered sequences derived from various types of amyloid sequences. AuNPs provide a nucleation site for MCAP assembly and colorimetric responses via the proteolytic activity of $M^{pro}$. Second, the homogeneity and stability of MCAP-AuNPs are necessary for highly reproducible high-throughput drug screening. We optimized the size and stability of MCAP-AuNPs to maximize the reproducibility of our platform. Specifically, we optimized the concentration of MCAP monomers to fabricate a homogeneously coated amyloid corona. These MCAP-AuNPs maintained stability in a high-salt buffer solution even after repeated freeze-thaw cycles, which is attributed to the amyloid hard corona. Third, our platform necessitates high sensitivity to accurately monitor the proteolytic activity of $M^{pro}$. We employed MCAP-AuNPs as a protease-sensitive nanomaterial to dynamically track the activity of $M^{pro}$ over time. Active $M^{pro}$ progressively cleaved the MCAP, resulting in the loss of the amyloid corona and aggregation of MCAP-AuNPs. These proteolytic reactions triggered colorimetric responses in MCAP-AuNPs, achieving a sensitivity that is remarkably 50 times lower than that of the commercialized FRET method. Furthermore, it exhibited selectivity only for $M^{pro}$. These results underscore the performance of our platform in monitoring $M^{pro}$'s proteolytic activity, rendering it optimal for assessing the efficacy of $M^{pro}$ inhibitors.

Finally, various drug candidates including four reagents (ebselen, hesperetin, leupeptin, and lopinavir) as $M^{pro}$ inhibitors were applied to our MCAP-AuNP-based drug screening platform. From the experiments, we demonstrated that this strategy with MCAP-AuNPs precisely identifies effective drugs that inhibit the proteolytic activity of $M^{pro}$. Furthermore, we successfully measured the $IC_{50}$, $V_{max}$, and $K_m$ values of each drug candidate using our drug screening platform, and these values were comparable to those reported previously[11,42,45,59–61]. From the pharmacokinetic results, we also determined the $K_i^{Exp}$ values and compared them to the $K_i^{MD}$ values to assess the discrepancy between experimental and simulated values. Although further validation, such as immunoassays or cellular assays, is required for drug candidates that tested positive, our platform can dramatically reduce the time and cost associated with the drug discovery process and by extent the subsequent validation steps[62]. In addition, this simple and rapid screening system can be applied to drug screening for COVID-X in the future because of the high genetic similarity of $M^{pro}$ within the coronavirus family.

In this study, we designed an $M^{pro}$ inhibitor screening platform based on colorimetric changes in MCAP-AuNP solutions. This rapid, label-free screening platform capable of real-time monitoring may provide a highly efficient and specific drug discovery system for COVID-19. This MCAP-AuNP-based strategy uses a bioengineered peptide, MCAP, which comprises an $M^{pro}$ cleavage site and an amyloid-forming sequence. This mimics the in vivo process wherein $M^{pro}$ cleaves the LQS sequence of polyproteins into nsps that are essential for viral replication. In addition, using the self-assembly property of the amyloid sequence, MCAP proteins can be coated on the surface of each AuNP to form an amyloid corona. The MCAP-AuNPs allowed the quantitative measurement of the proteolytic activity of $M^{pro}$ via the colorimetric response due to the intrinsic plasmonic properties of AuNPs. We also confirmed that this screening platform was suitable for $M^{pro}$ inhibitor screening by evaluating the inhibition efficacy of $M^{pro}$ inhibitors (e.g., ebselen, leupeptin, hesperetin, and lopinavir). We anticipate that our MCAP-AuNP-based platform can rapidly screen drug candidates in a chemical database to find antiviral therapeutics for COVID-19 and forthcoming COVID-X.

## Methods

### Reagents

Lyophilized SARS-CoV-2 main protease (M^pro) was purchased from Biosynth Carbosynth (UK). M^pro cleavage site-embedded amyloid peptides (MCAP; LQGNLQSNQQNY, MCAP2; LQGNLQANQQNY, MCAP3; LQGNLQGNQQNY, IAPP MCAP; SNLQSNFGAIL, Aβ MCAP1; KLLQSVFFAE, Aβ MCAP2; GGLQSVVIA) were commercially synthesized from Peptron (South Korea). Ebselen, hesperidin, hesperetin, lopinavir, leupeptin, chloroauric acid trihydrate (HAuCl$_4$ • 3H$_2$O), hydrochloric acid (HCl), thioflavin T (ThT), bovine serum albumin (BSA), immunoglobulin G (IgG), glucose, human serum albumin (HSA), trisodium citrate, and MCA-AVLQSGFR-Lys(Dnp)-Lys-NH2 trifluoroacetate were purchased from Sigma-Aldrich (USA). Distilled water (DW) and phosphate-buffered saline (PBS) were purchased from Gibco (USA).

### Preparation of engineered amyloid peptides (MCAP, MCAP2, MCAP3, IAPP MCAP, Aβ MCAP1, Aβ MCAP2, Repositioned MCAP) in vitro and fabrication of engineered amyloid fibrils

The monomers of each engineered amyloid peptide were purified as an acetate salt to prevent reacting to other agents. To aliquot the engineered amyloid peptides, lyophilized monomers were dissolved in DW. The monomer solutions (50 µL; 1 mg mL$^{-1}$) were distributed into 1.7-mL microcentrifuge tubes and stored at -20 °C in a freezer before the further experiment. To fabricate fibrils of each engineered amyloid peptides, 150 µL of DW (pH 2) was added to 50 µL of monomer solutions, and the mixture was incubated in a shaking incubator (Eppendorf, Germany) at 37 °C with shaking at 1000 Hz for 5 days.

### ThT fluorescence measurement for monitoring MCAP fibrillation

To investigate amyloid fibrillation kinetics, ThT fluorescence assays were performed using a microplate reader (Synergy H1 Multi-Mode Reader, BioTek, USA) at an excitation wavelength of 440 nm and an emission wavelength of 485 nm. First, 1 mg mL$^{-1}$ of MCAP solution was prepared by diluting the peptide in Milli-Q water. Subsequently, DW (pH 2) and a 1 mM ThT solution were added to the peptide solution until the final concentrations of the peptide and ThT were 250 µg mL$^{-1}$ and 20 µM, respectively. ThT intensity was recorded at steps of 30 min and performed at 37 °C for 99 h with continuous orbital shaking (807 CPM).

### ThT analysis of engineered amyloid fibrils

Before the ThT intensity measurement, each type of engineered amyloid fibrils was treated with M^pro for 4 h. After that, each 100 µM solution of engineered amyloid fibrils was treated with 20 µM of ThT molecules for 1 h. Finally, the ThT fluorescence intensities were measured by excitation at 444 nm and emission at 510 nm using a microplate reader (HIDEX, Japan).

### Synthesis of AuNPs

To remove chloroauric acid residues, a round beaker was soaked in aqua regia (a 1:3 mixture of HNO$_3$ and HCl) and rinsed with DW. The citrate reduction method was used to synthesize AuNPs. In detail, 2.5 mL of 38.8 mM HAuCl$_4$ solution was mixed with 45 mL of Millipore water in a round beaker, and the mixture was heated to 100 °C with stirring at 1,200 rpm. After boiling, 1 mL of 80 mM sodium citrate solution was added to the mixture, and the final solution was boiled again for 1 h under stirring at 1,200 rpm to make AuNP solution. The final AuNP solution was cooled to room temperature (25 °C) and stored at 4 °C. AuNP synthesis was confirmed by measuring their hydrodynamic diameter and polydispersity index (PDI) using Zetasizer Nano S90 (Malvern Instruments, UK).

### Synthesis of MCAP-AuNP

A 10:1 mixture of DW and PBS was added to the aliquoted MCAP to make a 0.1 mg mL$^{-1}$ MCAP solution. And then, for 1000 µL of AuNP solution, 960 µL of supernatant was removed via centrifugation for 20 min at 6720 × g. Next, 40 µL of the AuNP solution and 110 µL of DW were added to 50 µL of the MCAP solution. The final 200 µL solution was incubated for 24 h (37 °C with shaking at 1000 Hz) in a thermomixer (Eppendorf, Germany) to fabricate MCAP-AuNP solution. Successful MCAP-AuNP synthesis was confirmed using FE-TEM (JEM-2100F, Japan) imaging and hydrodynamic diameter measurements.

### Atomic force microscopy (AFM) analysis

Before topological conformation and height analysis using AFM, a silicon wafer was rinsed with a piranha solution (a 1:1 mixture of H$_2$SO$_4$ and H$_2$O$_2$). Fifty microliters of the MCAP monomer and fibril solution were deposited on the silicon wafer at room temperature for 20 min, washed with DW, and then dried for 12 h in a fume hood. AFM analysis was performed using an NX10 system (Park Systems, South Korea) with non-contact cantilever probes (NCHR, Park Systems, South Korea). AFM measurements were conducted in tapping mode at a scanning rate of 0.4-Hz and an image size of 5 × 5 µm. Image flattening and topological analysis were conducted using Park Systems' Smart Scan software.

### Circular dichroism (CD) analysis

Before the conformational analysis of MCAP fibrils using CD, MCAP fibrils were synthesized with varying incubation times. Briefly, 150 µL of DW (pH 2) was added to 50 µL aliquots of MCAP solution, and each mixture was incubated in a shaking incubator at 37 °C with 1000-Hz shaking for 12, 24, 36, 48, 60, and 96 h. Each sample was then deposited in a quartz glass cuvette (Aireka Cells, USA) with a 1 mm path length and 10 mm internal width. The CD spectra of each sample were measured using a J-815 system (Jasco, Japan), at a detection range of 190−300 nm and a scanning rate of 10 nm min$^{-1}$. The spectra had a resolution of 8 nm and were analyzed using CDTool software (Birkbeck College, UK).

### Fourier transform infrared (FT-IR) analysis

To perform conformation analysis of MCAP fibrils using FT-IR, 150 µL of DW (pH 2) was added to 50 µL of MCAP solution, and the mixture was incubated in a shaking incubator at 37 °C for 5 days to make MCAP fibrils. The MCAP fibril solution was then centrifuged at 20,000 × g for 1 h and the supernatant was removed. The resulting pellet was deposited on a silicon wafer and dried for 12 h in a fume hood. FT-IR spectra were measured using a Cary 630 FTIR Spectrometer (Agilent Technologies, USA) with a scanning range of 1600−1700 cm$^{-1}$. The spectra had a resolution of 4 nm and were analyzed using Agilent MicroLab software.

### X-ray photoelectron spectroscopy (XPS) analysis

The supernatants of the MCAP-AuNP and bare AuNP solutions were removed via centrifugation (20 min, 6720 × g). The pellets obtained after centrifuging each solution (<40 µL) were deposited on a silicon wafer and dried for 24 h in a fume hood. For XPS measurements, a K-alpha instrument (Thermo VG, UK) was used with monochromatic X-ray source (Al Kα line:1486.6 eV) at an ultra-high vacuum condition (4.8 × 10$^{-9}$ mbar). Elemental scans of Au and N peaks were detected using a pass energy of 40 eV, a step size of 0.1 eV, and scanning a range of 0−800 eV.

### UV-vis absorbance measurement

The UV-vis absorption spectra of each sample were measured using a spectrophotometer (PerkinElmer, USA), at a scan range of 400−800 nm, and a scan rate of 600 nm min$^{-1}$. For the US-vis absorbance spectra of AuNPs smaller than 20 nm, which have a dispersed state, the absorbance spectra peaks were detected at 525 nm ($A_{525}$). When the AuNPs aggregate, which leads to a color change in solution, the absorbance peak shifts near 650 nm ($A_{650}$). Thus, the degree of AuNP aggregation could be quantified by calculating the $A_{650}/A_{525}$ ratio.

## Freeze-thaw performance

One milliliter of MCAP-AuNP solution and bare AuNP solution in microcentrifuge tubes were frozen at −80 °C for 2 h. The frozen solutions were defrosted at room temperature for 2 h. This process was performed 8 times for repeated freeze-thaw tests. The hydrodynamic diameter of the MCAP-AuNP solution and bare AuNP solution before and after the freeze-thaw performance was measured using a Zetasizer. In addition, the UV-vis absorbance wavelength ranging from 400 to 800 nm of each solution before and after the freeze-thaw process was measured using a spectrophotometer. Each solution was photographed using a Galaxy Note 20.

## Molecular docking studies

To perform the molecular docking analysis, we followed a three-step protocol using three different software tools: Biovia Discovery Studio 2021 Client, UCSF Chimera version 1.14, and PyRx version 0.8.

Firstly, we retrieved the crystal structure of M$^{pro}$ (PDB ID: 6LU7) from the Research Collaboratory for Structural Bioinformatics Protein Data Bank (RCSB PDB) in.pdb format. We used Biovia Discovery Studio 2021 Client to remove unnecessary molecules, such as ions, inhibitors, and water molecules, and then loaded the processed structure of M$^{pro}$ into UCSF Chimera version 1.14. We performed Dock Prep to add hydrogen atoms and converted the processed structure of M$^{pro}$ into.pdbqt format using PyRx version 0.8.

Secondly, we retrieved the 3D structures of potential inhibitors of M$^{pro}$, ebselen (CID: 3194), hesperetin (CID: 72281), leupeptin (CID: 72429), and lopinavir (CID: 92727), from the PubChem database in.sdf format. We imported all four ligands into PyRx software, conducted energy minimization and geometrical confirmation using OpenBabel toolbox, and then converted the ligands into.pdbqt format.

Finally, we performed molecular docking analysis using AutoDock Vina inbuilt PyRx software version 0.8, based on the grid box approach. The grid box dimensions were set to $25 \times 25 \times 25$ Å along X-, Y- and Z-axes, with center coordinates of $X = -12.05$, $Y = 19.03$, $Z = 70.50$. To ensure accuracy, the catalytic active site residues, including His$^{41}$ and Cys$^{145}$, and substrate binding site residues were manually inspected and confirmed to be properly confined within the rectangular grid box during the grid generation process. We evaluated the binding free energies (kcal/mol) of the M$^{pro}$ and potential inhibitors complex after the docking analysis. In specific, the chain of 6LU7 was considered a rigid body and docked with flexible potential inhibitor ligands. Following the docking process, each ligand produced multiple docking poses, with up to nine poses based on root mean square deviation values. The first pose of each potential inhibitor, possessing the highest score, was selected for comparison of the docked ligand structures and binding affinities with M$^{pro}$. The docked complex files were then subjected to interaction studies, where we analyzed different types of interactions, such as covalent, carbon-hydrogen, hydrophobic interactions, and Van der Waals attractions using Biovia Discovery Studio.

## Kinetic analysis of M$^{pro}$ by using MCAP-AuNP

Before reaction with MCAP-AuNPs, various concentrations of M$^{pro}$ (0.925–18.5 nM) were dissolved in PBS for 20 min. Afterward, MCAP-AuNP solutions were added to each M$^{pro}$ solution to make a total volume of 1 mL. The mixtures of MCAP-AuNP and M$^{pro}$ solutions were then incubated at 37 °C for 1 h. After incubation, the UV-vis absorption spectra were measured using a spectrophotometer. The degree of particle aggregation ($A_{650}/A_{525}$) with respect to M$^{pro}$ concentration was represented using a sigmoidal dose–response curve.

## Monitoring M$^{pro}$ activity using FRET-based assay

Each 16 µM solution of MCA-AVLQSGFR-Lys(Dnp)-Lys-NH$_2$ trifluoroacetate (Sigma Aldrich, USA) substrates was reacted with various concentrations of M$^{pro}$ (0, 20, 74, 148, 296, 592, and 1184 nM) for 4 h.

Then, the fluorescence intensities were measured by excitation at 444 nm and emission at 510 nm using the microplate reader.

## Monitoring the efficacy of M$^{pro}$ inhibitors using MCAP-AuNPs

M$^{pro}$ (50 ng) was added to M$^{pro}$ inhibitors at various concentrations (ebselen: 0.009 µM to 9.25 µM, hesperetin: 4.625 to 185 µM, lopinavir: 0.925 to 92.5 µM, leupeptin: 0.094 to 281.25 µM, and hesperidin: 11.563 to 370 µM) dissolved in PBS (1.5% DMSO), and the total volume of the solution was 800 µL. These solutions were incubated at room temperature for 20 min and then filtered using a 200-µm-pore PVDF syringe filter (Biopil, China) to remove the undissolved drugs. Then, 600 µL of the filtered solution was added to 200 µL of MCAP-AuNP solution and incubated at 37 °C for 1 h. After incubation, the UV-vis absorption spectra of the solutions were measured using a spectrophotometer. The degree of particle aggregation was represented by the relative UV-vis absorbance ratio $A_{650}/A_{525}$.

## Monitoring the enzymatic activity of M$^{pro}$ using MCAP-AuNP

M$^{pro}$ (12 µg) was added to M$^{pro}$ inhibitors (ebselen, hesperidin, hesperetin, lopinavir, and leupeptin) at various concentrations dissolved in PBS (10% DMSO), and the total volume of the solution was 500 µL. These solutions were then incubated at room temperature for 20 min and then filtered using a 200-µm-pore PVDF syringe filter to remove the undissolved drugs. Afterward, 70 µL of the filtered solutions were added into 1.88 µM, 3.7 µM, 5.66 µM, 7.55 µM, 9.43 µM, and 11.32 µM of MCAP solutions with a final volume of 220 µL. Using a microplate reader (Molecular Device, USA), inhibitor-controlled M$^{pro}$ activity with respect to MCAP-AuNP concentration was calculated using the relative UV-vis absorbance ($A_{650}/A_{525}$). Enzymatic activities were calculated using the Michaelis-Menten equation.

## Reporting summary

Further information on research design is available in the Nature Portfolio Reporting Summary linked to this article.

# Data availability

All data supporting this research are available within the article and its Supplementary Information files or from the corresponding author upon request. Source data are provided in this paper. Source data file includes raw data underlying the respective main text (Figs. 1, 2, 3, and 5) and Supplementary Information (Supplementary Figs. S3, 4, 5, 6, 7, 8, 9, 10, 11, 12, 13, 14, 15, 16, 17, 18, 19, 20, 21, and 22). The data for molecular docking are from the Protein Data Bank (PDB accession code: 6LU7) and the PubChem compound database under accession codes: 3194, 72281, 72429, and 92727. To visualize the origins of the amyloid sequence and M$^{pro}$ cleavage sequence in Fig. 1b, the crystal structures of the yeast prion protein Sup35 (PDB accession code: 1R5B) and 2019-nCoV nsp7-nsp8c complex (PDB accession code: 6M5I) were used. Input and output files of AutoDock Vina calculations are provided in Supplementary Data 1. Source data are provided in this paper.

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

## Acknowledgements

This work was supported by a National Research Foundation of Korea (NRF) grant funded by the Korean Government (MSIP) (NRF-2022R1A2C1091756, NRF-2020R1A2C2102262, NRF-2021R1A2C2004760 and NRF-2022R1A6A3A03066467), BK21 FOUR Institute of Precision Public Health, Health Fellowship Foundation, and NRF under project BK21 FOUR operated by the Center for Teaching and Learning of Korea University. This work was also supported by the Korea Health Industry Development Institute (KHIDI) grant HU21C0053 and the Bio & Medical Technology Development Program of the National Research Foundation funded by the Korean government (MSIT) (No. 2023M3E5E3080743).

## Author contributions

D. Lee, H. G. Jung, D. Park, J. H. Lee and D. S. Yoon conceived and designed the study. D. Lee, H. G. and J. Bang conducted the analysis of MCAP and overall experiments. D. Park performed molecular structure analysis. D. Y. Cheong, G. Lee and Y. H. Kim supported the ThT assay and AFM analysis. J. W. Jang, Y. Kim, S. Lee, S. W. Lee and J. H. Hong supported data analysis. D. Lee, H. G. Jung, D. Park, K. S. Hwang, J. H. Lee, D. S. Yoon drafted the manuscript. All authors discussed the results and commented on the manuscript.

## Competing interests

The authors declare no competing interests.
