## [Peer Review File · Nature Communications]

Bioengineered amyloid peptide for rapid screening of inhibitors against main protease of SARS-CoV-2REVIEWER COMMENTS

Reviewer #1 (Remarks to the Author):

The authors devised a method to screen protease inhibitors for SARS-CoV-2, utilizing bioengineered amyloids coated on gold nanoparticles. The activity of protease can be monitored by the change in SERS signals, where the amyloid cleavage destabilizes the suspension of the individual nanoparticles and promotes their aggregation. It seems that overall conclusion of the manuscript is well supported by the experimental data, and most of background studies have been properly referenced. Thus, I recommend the publication of the manuscript after considering the following issues:

1. I just wonder that the 3D geometry of the amyloids attached on the Au NPs would be different from those in suspension, and the protease activity can be sterically hindered by the large size of the protease protein limiting the access to the surface of Au NPs. I guess, in terms of dynamic chemical equilibrium, that the amyloids can be reversibly detached from Au NPs, facilitating the reaction with the protease, which needs to be discussed in the manuscript.
2. Enzymatic degradation of the amyloids were confirmed by SEM, ThT assay, CD spectra, and AFM, but I would like to recommend more molecular analyses such as mass spectroscopy and FT-IR. In addition, the authors claim that the length of the MCAP fibril is shorter than the original amyloid fibril based on the height information in AFM (Fig. S1-S2), but I'm not sure if the AFM height is correlating with the length of peptides. (This is the same for Fig 1e-f.)
3. It seems that the number of MCAP-AuNPs in Fig 2b is too small, and the size distribution needs to be confirmed from a larger set of the NPs.
4. The inhibition ratio calculated from A650/A525 needs to be cross-checked. In my opinion, the inhibition ratio(%) should be $(A535/A650)/\text{Maximum}(A525/A650) \times 100$ in case that the inhibition ratio is supposed to increase with color change from 650 nm to 525 nm.

Reviewer #2 (Remarks to the Author):

I'm aware that Nature Comms has copy editors, but I can't restrain myself from pointing out even very small errors in writing. Below I will pass over the text and introduce concerns that I have in order of reading, both large and small, ending with a summary of my opinion of the draft.

ABSTRACT:

"These results demonstrated that our MCAP-AuNP-based platform great potential to discover Mpro inhibitors and may accelerate the development of therapeutics against COVID-19."

>>This sentence lacks a "has"

INTRODUCTION:

On sentence one I feel like I am not going to get along with this paper:

"pandemic that infected every age group with a rapid and uncontrollable spread"

>>pandemic that spread rapidly across all age groups

The phrasing of the above quoted text is awkward, and also the claim that SARS-CoV2 was uncontrollable is subject to debate. It wasn't very well controlled (except in some fortunate countries such as New Zealand) but that doesn't mean that it couldn't have been controlled.

"over 682m are currently infected with SARS-CoV-2"

This text needs to be sourced, and clarified with timepoint at which this value applied. Hopefully it won't always be true that 682m have covid.

Paragraph two: "The main protease"

It is just a suggestion, but maybe mention in one line the analogy here with HIV-I protease, which performs a similar function and has been an immensely important antiviral drug target.

"%, whereas their genomes only have 82%"

>>%, whereas their genomes overall have only 82%

"Therefore, considering the self-assembly properties of amyloid proteins, we speculated that if"
The text beginning with this line is the important part of the introduction, explaining why the authors did what they did. In broad terms (details are explained in the next para), what is the advantage of making NP-conjugated amyloid?

Panel (b) of figure 1 does some of the work of introducing the concept of the paper, but there needs to be a new figure, "figure zero", which gives a conceptual explanation of the function of the system devised. This figure should be part of the Introduction. As far as I understand it at the end of reading the Introduction, then skipping ahead and reading backwards and forwards a little bit, the logic here is:

(1) conjugate prion domains NQQNY and LQGN to the proteolysis target motif for MPRO, which is LQS.

(2) attach these constructs (called MCAPs) to Au NPs, making MCAP-AuNPs

(3) If MPRO is effective, then it will prevent/reverse self-assembly of the MCAP-AuNPs, which can be read out optically

(4) If some small molecule suppresses MPRO, then the MCAP-AuNPs will still self assemble, congratulations you have discovered a drug lead

I feel like this idea should be clear from the abstract, and very clear (with a figure) after reading the Introduction. It wasn't, I had to work to understand what is going on.

Another major concern I have with the introduction is that there is no reference to the multiple ongoing or existing therapeutics targeting MPRO. A review is here:

<https://doi.org/10.1002/mco2.151> [Hu et al], the authors should read this and other literature and acknowledge at least a cross section of the important work on MPRO.

My final point to make here is that SARS-CoV2 is rich in amyloidogenic proteins. The authors should discuss this point (with sources), and address its relevance or otherwise, in particular addressing the question (as far as possible given current literature) of whether self-assembled substrates are what MPRO works on in vivo.

RESULTS:

Paragraph one mentions only ACE2 as an entry point. There are many (see eg:

<https://www.nature.com/articles/s41580-021-00418-x>) and this should be noted. Discussing only ACE2 is a

dangerous oversimplification.

"considerably compared to that of the fibrils"

>>Did you mean: "considerable compared to that of the fibrils"?

"they forms the"

>>they formed

Getting into the results, this section of the paper is substantial and cleanly written.

The figures don't all look great, I am going to assume that higher resolution or vector figures will

be provided for the resubmission.

Text in the figures is not of consistent size, and is overall too small.

DISCUSSION:

Short and to the point, as I like it.

REFERENCES:

There are multiple glitches in the references and missing citations, obviously nobody has given any attention to the bibliography before submission.

Referee summary:

This paper presents an interesting technology for screening SARS-CoV2 Main Protease inhibitors.

The technical presentation of the material is clear. The placing of the work in context has substantial room for improvement, and is careless in that the bibliography has five unresolved citations that I counted. Important results and ongoing trends in the literature are missed. This work may be important, and it may be original, but unless the authors do the work of placing it in context this is impossible to state with confidence.

The referee is aware that multiple protease-sensitive nanoparticle systems exist (example: "Protease-Sensitive Nanomaterials...", PMC5445504) and the present work needs absolutely to be put in context with these other systems. The search term "protease-sensitive nanomaterial" needs to be in the text such that the final paper will be discoverable in context with other similar work.

Reviewer #3 (Remarks to the Author):

Authors have presented the colorimetric based screening platform to identify Mpro protease inhibitors for SARS-CoV-2. They have systematically engineered the Mpro cleavage site (LQS) into amyloid-forming sequences from prion protein (GNNQQY) without compromising the amyloidogenic nature (MCAP). The peptide was fabricated onto AuNPs to generate MCAP-AuNPs that was employed to develop the colorimetric assay. The surface decorated peptide prone for degradation in the presence of Mpro destabilizes the AuNPs that results in the agglomeration of AuNPs and readout as color change. The colorimetric change was successfully exploited to screen the drug candidates that inhibit Mpro. The work seems interesting in terms of drug discovery whereas lacks novelty of the approach. There are a couple of control experiments that need to be performed to substantiate the claims in the work. The manuscript can be accepted after addressing the major concerns enlisted below.

1. LQS sequence was chosen for Mpro recognition unit among three possible combinations from L-Q↓(S, A, G). What is the rationale for this biased selection. Authors need to engineer rest and check for their Mpro selectivity and amyloidogenicity.
2. There are numerous amyloid peptides. What is the rationale to choose GNNQQY. The authors needs to include one more or more control amyloid to show the superior performance of the current one in the developed assay.
3. The authors have demonstrated the amyloid fibril degrading ability of Mpro. The fibrils are known to assemble with strong hydrophobic and hydrogen bonding interaction results in embedding the core structure of peptides in the interior of the fibrils. How is the enzyme active site accessing the cleavage sequence in the core of peptide in fibrils structure? The enzyme active sites are very specific with precise special arrangements that allow them to fit the target polypeptide sequence. There is a need to address this by designing some experiments or computation.

4. The authors describe MCAP acquire β -sheet on the AuNPs driving the nucleation. The β -conformation on AuNPs which needs to be substantiated by CD spectroscopy. Because in the local environment on the surface of charged AuNPs, behaviour of the peptide is uncertain and unclear.
5. The DLS experiments indicates the reduction of negative charged AuNPs. The studies show 98% coating of AuNPs with MCAP. Still the negative charge persists to a good extent. Both are contradicting. Is the coating percentage being overestimated?
6. For stability study, freeze thaw was used. The temperature is another important parameter in a realistic manner. Are the MCAP-AuNPs stable at higher temperature or at AuNPs can be employed for Mpro activity detection. How many freeze thaw cycles can they sustain without losing activity.
7. The authors demonstrated the ability of MCAP-AuNPs to detect Mpro by aggregation of AuNPs. How to rule out the effect of Mpro to induce the aggregation by noncovalent interaction with MCAP-AuNPs. Suitable controls like BSA need to be used to check the ability of protein to induce MCAP-AuNPs aggregation. Vice versa, the effect of MCAP-AuNPs on the activity of Mpro needs to be ruled out by including known substrate for Mpro.
8. In the drug screening assay, the possibility of drugs interacting with MCAP-AuNPs by non-covalent interactions preventing Mpro to degrade MCAP-AuNPs exists. Need to include appropriate control to rule out this possibility. How suitable this assay for HTS need to be discussed.

Response Letter

We hereby resubmit our revised manuscript (Manuscript ID: NCOMMS-23-20825A). We have responded to the comments of each reviewer point-by-point. All revisions and corrections in our revised manuscript are shown in blue.

Reviewer #1 (Remarks to the Author):

The authors devised a method to screen protease inhibitors for SARS-CoV-2, utilizing bioengineered amyloids coated on gold nanoparticles. The activity of protease can be monitored by the change in SERS signals, where the amyloid cleavage destabilizes the suspension of the individual nanoparticles and promotes their aggregation. It seems that overall conclusion of the manuscript is well supported by the experimental data, and most of background studies have been properly referenced. Thus, I recommend the publication of the manuscript after considering the following issues:

Our response: We appreciate your insightful comments and have revised our manuscript accordingly.

1. I just wonder that the 3D geometry of the amyloids attached on the AuNPs would be different from those in suspension, and the protease activity can be sterically hindered by the large size of the protease protein limiting the access to the surface of AuNPs. I guess, in terms of dynamic chemical equilibrium, that the amyloids can be reversibly detached from AuNPs, facilitating the reaction with the protease, which needs to be discussed in the manuscript.

Our response: Thank you for your valuable comment. Based on your comment, we conducted measurements of MCAP concentration in the supernatant solutions before (Supernatant A) and after the centrifugation (Supernatant B) using BCA assay (Fig. R1a). MCAP concentration in Supernatant A was found to be 68.7% compared to the initial MCAP concentration. In contrast, the MCAP concentration of the supernatant B (after centrifugation) was negligible. This indicates that the MCAP was formed as a hard amyloid corona and hardly detached from AuNP. In addition, we investigated the potential impact of free MCAPs on the reactivity with M^{pro} . As shown in Fig. R1b, the dose-response curves were observed with M^{pro} in the presence and absence of free MCAPs (the samples with/without centrifugation). Herein, free MCAPs represent free MCAP aggregates existing in suspension, not on AuNPs. The half-maximal effective concentration (EC_{50}) was slightly shifted from 3.5 nM to 2.1 nM in the presence of free MCAPs. These results suggest that free MCAPs can act as additional substrates to M^{pro} , which slightly changes the apparent reactivity between MCAP-AuNP and M^{pro} . Collectively, these results indicate that the MCAP is irreversibly bound to the surface of AuNPs. This finding is consistent with our previous studies [*Nat. Commun.*, 12(1), 639 (2021), *ACS Appl. Mater. Interfaces*, 15(2), 2538-2551 (2023)]. As per your suggestion, we have now discussed the reaction of free MCAP aggregates for M^{pro} in the revised manuscript (page 14, lines 2 to 11) and supporting information (Supplementary Fig. S20).

Figure R1. (a) Relative MCAP concentrations of the MCAP-AuNP solutions before (Supernatant A) and after purification (Supernatant B) compared to the initial MCAP concentration (Initial MCAP). (b) Sigmoidal dose-response curves as a function of M^{pro} concentration for the MCAP-AuNP solution in

the presence and absence of free MCAPs.

2. Enzymatic degradation of the amyloids were confirmed by SEM, ThT assay, CD spectra, and AFM, but I would like to recommend more molecular analyses such as mass spectroscopy and FT-IR. In addition, the authors claim that the length of the MCAP fibril is shorter than the original amyloid fibril based on the height information in AFM (Fig. S1-S2), but I'm not sure if the AFM height is correlating with the length of peptides. (This is the same for Fig 1e-f.)

Our response: We appreciate your insightful comment. Accordingly, we performed molecular analyses including Fourier-transform infrared (FT-IR) and circular dichroism (CD) analysis in Fig. R2. The FT-IR analysis showed the absorbance peak of MCAP fibrils to be near 1620 cm^{-1} wavelength, representing the beta-sheet structure of MCAP fibrils [*Protein science*, 13(12), 3314-3321 (2004)] (Fig. R2a). The peak of MCAP fibrils was notably reduced in the presence of M^{Pro} , indicating the proteolytic activity of M^{Pro} toward MCAP fibrils. In addition, the M^{Pro} decreased the peak of the beta-sheet structure (190 to 220 nm) of CD spectra [*Nature protocols*, 1(6), 2876-2890 (2006)], providing further evidence for the enzymatic degradation of MCAP fibrils (Fig. R2b). This additional information has been incorporated into our revised manuscript (page 8, lines 2 to 5) and supporting information (Supplementary Fig. S4).

Figure R2. (a) FT-IR analyses of the MCAP fibrils and the M^{Pro} -treated MCAP fibrils. (b) CD spectra of the MCAP fibrils and the M^{Pro} -treated MCAP fibrils.

To assess how the insertion of MCAP cleavage sites affects amyloidosis behavior, we compared the persistence length and height of MCAP fibrils with those of original amyloid fibrils (GNNQQNY) [*Angew. Chem. Int. Ed.* 55(37), 11242-11246 (2016)]. The persistence length was measured by Image J software from the TEM images and the height was measured by NX10 software from the AFM images. The persistence length and height of MCAP fibrils were measured as $229.95 \pm 94.22\text{ nm}$ and $2.64 \pm 0.74\text{ nm}$ (Fig. S2 and Fig. S3d-f), respectively. Our analysis revealed a considerable decrease in both the height and length of MCAP fibrils compared to the original amyloid fibrils (length: $1.9 \pm 1.3\text{ }\mu\text{m}$ and height: $43 \pm 24\text{ nm}$) due to the introduction of M^{Pro} cleavage sites. Regardless of the decrease in height and length, the MCAP retained excellent amyloidosis behavior enough to form amyloid corona on AuNP. Further clarification on this observation has been provided in our manuscript (page 7, lines 4 to 10).

3. It seems that the number of MCAP-AuNPs in Fig 2b is too small, and the size distribution needs to be confirmed from a larger set of the NPs.

Our response: We appreciate your comment and consider it valuable for improving our manuscript. Accordingly, we acquired the transmission electron microscopy (TEM) image with a larger number of

the MCAP-AuNPs staining by uranyl acetate at various magnifications (Fig. R3a to R3c). The average diameter of MCAP-AuNPs is measured as 22.47 ± 1.90 nm (Fig. R3d). We added this information to our revised manuscript (page 10, lines 16) and supporting information (Supplementary Fig. S14).

Figure R3. (a) to (c) TEM images of the MCAP-AuNPs. The size of white scale bar is 50 nm. (d) Size distribution of the MCAP-AuNPs. The diameter of each MCAP-AuNP was measured by Image J software in TEM images ($n=31$).

4. The inhibition ratio calculated from A_{650}/A_{525} needs to be cross-checked. In my opinion, the inhibition ratio(%) should be $(A_{535}/A_{650})/\text{Maximum}(A_{525}/A_{650}) \times 100$ in case that the inhibition ratio is supposed to increase with color change from 650 nm to 525 nm.

Our response: We appreciate your careful reviewing of the manuscript. The values of A_{650}/A_{525} of MCAP-AuNP, depending on drug concentration, are presented in Fig. R4. Following for your comments, we revised the inhibition ratio (%) to $(A_{525}/A_{650})/\text{Maximum}(A_{525}/A_{650}) \times 100$ in our revised manuscript (page 16, line 22 to page 17, line 1).

Figure R4. The values of A_{650}/A_{525} of the MCAP-AuNP reaction with M^{Pro} depending on inhibitor concentration. Each plot was fitted by the sigmoidal dose–response curve as a function of inhibitor concentration.

Reviewer #2 (Remarks to the Author):

I'm aware that Nature Comms has copy editors, but I can't restrain myself from pointing out even very small errors in writing. Below I will pass over the text and introduce concerns that I have in order of reading, both large and small, ending with a summary of my opinion of the draft.

Our response: We are grateful for your detailed reviewing and insightful suggestions. Your comments have assisted us considerably in improving the overall quality of the manuscript. We have provided detailed responses for each comment below.

ABSTRACT:

1. "These results demonstrated that our MCAP-AuNP-based platform great potential to discover Mpro inhibitors and may accelerate the development of therapeutics against COVID-19."

>>This sentence lacks a "has"

Our response: Thank you for pointing this out. We have revised this sentence as follows:

"These results demonstrated that our MCAP-AuNP-based platform has great potential for discovering M^{pro} inhibitors and may accelerate the development of therapeutics against COVID-19."

INTRODUCTION:

2. On sentence one I feel like I am not going to get along with this paper:

"pandemic that infected every age group with a rapid and uncontrollable spread"

>>pandemic that spread rapidly across all age groups

The phrasing of the above quoted text is awkward, and also the claim that SARS-CoV2 was uncontrollable is subject to debate. It wasn't very well controlled (except in some fortunate countries such as New Zealand) but that doesn't mean that it couldn't have been controlled.

Our response: We acknowledge that our description may have been misleading. Accordingly, the sentence has been revised as follows:

"The emergence of severe acute respiratory syndrome coronavirus-2 (SARS-CoV-2) has caused a global pandemic. The virus has exhibited a high rate of infectivity and the ability to spread rapidly across all age groups."

3. "over 682m are currently infected with SARS-CoV-2"

This text needs to be sourced, and clarified with timepoint at which this value applied. Hopefully it won't always be true that 682m have covid.

Our response: The sentence has been revised to clearly present the timepoint as per your recommendation, and the source of the data has been cited in the references.

"According to the World Health Organization (WHO), as of November 2023, there have been over 771 million confirmed infections and over 6.9 million cumulative deaths caused by SARS-CoV-2."

4. Paragraph two: "The main protease"

It is just a suggestion, but maybe mention in one line the analogy here with HIV-I protease, which performs a similar function and has been an immensely important antiviral drug target.

Our response: We fully agree with your comment and appreciate the suggestion. Accordingly, we have added the following description of HIV-1 and HCV related proteases to the introduction (page 3, lines 15 to 19; page 3, line 24 to page 4, line 1).

“Viral proteolytic enzymes, including human immunodeficiency virus-1 (HIV-1) and hepatitis C virus (HCV) NS3/4A proteases, play essential roles in viral proliferation and assembly, thereby making them promising potential therapeutic targets. Among these, the main protease (M^{pro}), a cysteine protease containing a His⁴¹–Cys¹⁴⁵ catalytic dyad, is an indispensable enzyme for SARS-CoV-2 replication and proliferation. The M^{pro} -mediated proteolytic post-processing of the SARS-CoV-2 replicase polyprotein, which cleaves at least 11 conserved sites, is crucial for viral assembly and maturation. For example, M^{pro} generates amyloidogenic proteins (e.g., spike protein and non-structural protein 11) in SARS-CoV-2 proteosomes with multiple aggregation-prone regions, facilitating viral self-assembly. Considering the proven success of protease inhibitors in treating HIV-1 and HCV infections, the strategies that specifically target M^{pro} possess significant potential to thwart viral proliferation.”

5. "%, whereas their genomes only have 82%"

>>%, whereas their genomes overall have only 82%

Our response: We appreciate your meticulous reviewing of the manuscript. We have revised this part as per your suggestion.

6. "Therefore, considering the self-assembly properties of amyloid proteins, we speculated that if"

The text beginning with this line is the important part of the introduction, explaining why the authors did what they did. In broad terms (details are explained in the next para), what is the advantage of making NP-conjugated amyloid?

Our response: Thank you for your valuable comment. Our drug screening strategy, based on MCAP-AuNPs, offers significant advantages. Firstly, AuNPs act as catalytic reactors, facilitating the aggregation of amyloid-forming proteins by providing a nucleation sites [*ACS nano*, 7(7), 6268-6277 (2013)], leveraging the fabrication of MCAP-AuNPs without additional chemical functionalization. In addition, our strategy utilizes plasmonic nanoparticles as optical reporters, allowing for the precise and quantitative monitoring of sequence-specific degradation by M^{pro} without any chemical dyes or antibodies. Based on these properties of AuNPs, our strategy allows for monitoring the proteolytic activity of M^{pro} with 10 times lower concentration (Fig.3F) compared to those required for the fluorescence resonance energy transfer (FRET) method [*Nature*, 582(7811), 289-293 (2020)].

Capitalizing on these advantages, our MCAP-AuNP-based platform can rapidly screen drug candidates, potentially accelerating the identification of effective antiviral therapeutics for COVID-19. We have revised the sentence in accordance with your comments as follows (page 5, lines 10 to 15).

“Therefore, considering the self-assembly properties of amyloid proteins, we speculated that a hybrid nanocomposite for drug screening can be synthesized with a combination of AuNPs and an amyloid peptide containing both M^{pro} cleavage and amyloid-forming sequences. By leveraging both the catalytic activity and the localized surface plasmon resonance (LSPR) property of AuNPs, this hybrid nanocomposite is applicable to a high-throughput screening platform for M^{pro} inhibitor candidates.”

7. Panel (b) of figure 1 does some of the work of introducing the concept of the paper, but there needs to be a new figure, "figure zero", which gives a conceptual explanation of the function of the system devised. This figure should be part of the Introduction. As far as I understand it at the end of reading the Introduction, then skipping ahead and reading backwards and forwards a little bit, the logic here is:

- (1) conjugate prion domains NQQNY and LQGN to the proteolysis target motif for MPRO, which is LQS.
- (2) attach these constructs (called MCAPs) to Au NPs, making MCAP-AuNPs
- (3) If MPRO is effective, then it will prevent/reverse self-assembly of the MCAP-AuNPs, which can be read out optically
- (4) If some small molecule suppresses MPRO, then the MCAP-AuNPs will still self assemble, congratulations you have discovered a drug lead

I feel like this idea should be clear from the abstract, and very clear (with a figure) after reading the Introduction. It wasn't, I had to work to understand what is going on.

Our response: We completely agree with your comment that a new figure is needed for a clearer representation of the conceptual explanation of our drug screening platform. To avoid any potential confusion for the reader, we have added Supplementary Fig. S1, which shows the workflow of the MCAP-AuNP-based drug screening platform.

1. Fabrication of MCAP-AuNPs

2. MCAP-AuNP-based screening platform for M^{Pro} inhibitors

Supplementary Fig. S1. Schematic illustration of the MCAP-AuNP-based drug screening platform.

8. The major concern I have with the introduction is that there is no reference to the multiple ongoing or existing therapeutics targeting MPRO. A review is here: <https://doi.org/10.1002/mco2.151> [Hu et al], the authors should read this and other literature and acknowledge at least a cross section of the important work on MPRO.

Our response: We appreciate your comment. We have now included a description in the introduction regarding M^{pro} inhibitors that have been approved by the FDA or are in clinical trials for the treatment of COVID-19. The revised text is as follows: (page 4, lines 8 to 12)

“For instance, Paxlovid, an oral therapeutic that combines an M^{pro} inhibitor (PF-07321332) with ritonavir, has garnered FDA approval for the treatment of moderate to severe COVID-19 cases. Moreover, compounds like Pfizer's PF-07304814 and Simcere's SIM0417, designed to inhibit M^{pro}, are currently undergoing clinical trials as potential oral treatments against SARS-CoV-2.”

9. My final point to make here is that SARS-CoV2 is rich in amyloidogenic proteins. The authors should discuss this point (with sources), and address its relevance or otherwise, in particular addressing the question (as far as possible given current literature) of whether self-assembled substrates are what MPRO works on *in vivo*.

Our response: We appreciate your insightful comments regarding the abundance of amyloidogenic protein in SARS-CoV-2. In particular, M^{pro} generates amyloidogenic proteins with multiple aggregation-prone regions in SARS-CoV-2 proteosomes, facilitating viral self-assembly [*Nature Communications*, 14(1), 945, (2023)]. For example, spike protein and non-structural protein 11 are formed through self-assembly of peptide fragments after M^{pro} cleavage. Accordingly, M^{pro} does not work on spike protein and non-structural protein 11, as they are final products by M^{pro} cleavage.

Our self-assembled MCAP-AuNP platform utilizes a bioengineered sequence (MCAP) which is not a naturally occurring amyloid protein *in vivo*. To avoid any potential confusion for the reader, we clarified in the Result section that the MCAP sequence was designed by combining a prion-derived amyloidogenic sequence with the M^{pro}-cleavage site.

Considering the importance of exploring the broader context of amyloidogenic proteins within SARS-CoV-2 proteome, we have incorporated a discussion on the prevalence of amyloid-forming peptides in the revised manuscript (page 3, lines 21 to 23).

“The M^{pro}-mediated proteolytic post-processing of the SARS-CoV-2 replicase polyprotein, which cleaves at least 11 conserved sites, is crucial for viral assembly and maturation. For example, M^{pro} generates amyloidogenic proteins (e.g., spike protein and non-structural protein 11) in SARS-CoV-2 proteosomes with multiple aggregation-prone regions, facilitating viral self-assembly.”

RESULTS:

10. Paragraph one mentions only ACE2 as an entry point. There are many (see eg: <https://www.nature.com/articles/s41580-021-00418-x>) and this should be noted. Discussing only ACE2 is a dangerous oversimplification.

Our response: We fully agree with your comments and acknowledge the aspect we had overlooked. Accordingly, we have now revised the beginning of the first paragraph as follows.

“Fig. 1a shows the endosomal entry and replication process of SARS-CoV-2. This process is initiated by binding between human angiotensin-converting enzyme-2 (hACE-2) receptor and spike protein (S protein) of SARS-CoV-2. Following these interactions, the S protein undergoes proteolytic cleavage by cathepsin L, leading to the conversion of the protein into a metastable state, subsequently triggering the fusion of the host cell membrane with the virus.”

11. "considerably compared to that of the fibrils"

>>Did you mean: "considerable compared to that of the fibrils"?

"they forms the"

>>they formed

Our response: We appreciate your meticulous reviewing of the manuscript and have revised this text accordingly.

12. Getting into the results, this section of the paper is substantial and cleanly written.

The figures don't all look great, I am going to assume that higher resolution or vector figures will be provided for the resubmission. Text in the figures is not of consistent size and is overall too small.

Our response: Thank you for your comments. As suggested, the text size in the figures has been adjusted, and high-resolution images have been included.

DISCUSSION:

Short and to the point, as I like it.

REFERENCES:

13. There are multiple glitches in the references and missing citations, obviously nobody has given any attention to the bibliography before submission.

Our response: Thank you for pointing this out. We have checked all the references before resubmitting this revised manuscript.

14. Referee summary:

This paper presents an interesting technology for screening SARS-CoV2 Main Protease inhibitors.

The technical presentation of the material is clear. The placing of the work in context has substantial room for improvement and is careless in that the bibliography has five unresolved citations that I counted. Important results and ongoing trends in the literature are missed. This work may be important, and it may be original, but unless the authors do the work of placing it in context this is impossible to state with confidence.

The referee is aware that multiple protease-sensitive nanoparticle systems exist (example: "Protease-Sensitive Nanomaterials...", PMC5445504) and the present work needs absolutely to be put in context with these other systems. The search term "**protease-sensitive nanomaterial**" needs to be in the text such that the final paper will be discoverable in context with other similar work.

Our response: We are sincerely grateful for your insightful comments, as they have highlighted important points that we had previously overlooked. Following your recommendation, we have added a sentence in the Introduction section (page 5, lines 4 to 6; page 5, lines 21 to 22) to ensure that our paper is identifiable in searches pertaining to 'protease-sensitive nanomaterial' as follows.

"In particular, there has been some progress in developing protease-responsive nanomaterials that

capitalize on the degradation of amyloid aggregates or fibrils.” (page 5, line 3-5)

“By combining MCAPs and AuNPs, we fabricated novel protease-sensitive nanocomposite, termed MCAP-AuNPs, wherein AuNPs were coated with the MCAP amyloid corona.” (page 5, line 21-22)

Reviewer #3 (Remarks to the Author):

Authors have presented the colorimetric based screening platform to identify Mpro protease inhibitors for SARS-CoV-2. They have systematically engineered the Mpro cleavage site (LQS) into amyloid-forming sequences from prion protein (GNNQQY) without compromising the amyloidogenic nature (MCAP). The peptide was fabricated onto AuNPs to generate MCAP-AuNPs that was employed to develop the colorimetric assay. The surface decorated peptide prone for degradation in the presence of Mpro destabilizes the AuNPs that results in the agglomeration of AuNPs and readout as color change. The colorimetric change was successfully exploited to screen the drug candidates that inhibit Mpro. The work seems interesting in terms of drug discovery whereas lacks novelty of the approach. There are a couple of control experiments that need to be performed to substantiate the claims in the work. The manuscript can be accepted after addressing the major concerns enlisted below.

Our response: We really appreciate your insightful comments. As per your feedback, we have revised the text to enhance the context and overall quality of our manuscript.

1. LQS sequence was chosen for Mpro recognition unit among three possible combinations from L-Q↓(S, A, G). What is the rationale for this biased selection. Authors need to engineer rest and check for their Mpro selectivity and amyloidogenicity.

Our response: Thank you for your valuable comment. We have designed our MCAP sequence by inserting the LQS sequence into the amyloid-forming sequence as L-Q↓(S, A, G) is recognized to undergo cleavage by M^{pro} [*Nature*, 582(7811), 289-293 (2020)]. To assess the rest of the engineered sequences for M^{pro} selectivity and amyloidogenicity, we newly designed engineered amyloid sequences, where the LQS within the MCAP sequence (LQGNLQSNQQNY) was changed to LQA (LQGNLQANQQNY, MCAP2) and LQG (LQGNLQGNQQNY, MCAP3) (Table. R1). To confirm the amyloidogenicity of the MCAP2 and MCAP3, we conducted atomic force microscopy (AFM) analysis, transmission electron microscopy (TEM) imaging, and circular dichroism (CD) analysis (Fig. R5, and Fig. R6).

Name	Originated protein	Originated sequence	Engineered sequence
MCAP	Prion protein	GNNQQNY	LQGNLQSNQQNY
MCAP2	Prion protein	GNNQQNY	LQGNLQANQQNY
MCAP3	Prion protein	GNNQQNY	LQGNLQGNQQNY

Table. R1. Engineered amyloid sequences based on prion protein-derived amyloid-forming sequence (GNNQQNY) and M^{pro}-cleavage sequences (L-Q↓(S, A, G)).

Fig. R5a, b showed the topology of MCAP2 fibrils with line profiles analyzed by AFM. We further validated the amyloid-forming property of MCAP2 fibrils through TEM imaging in Fig R5c, d. The beta-sheet structure of MCAP2 fibrils was observed by CD analysis in Fig. R5e with peaks near 210 cm⁻¹ and 190 cm⁻¹ [*Nature protocols*, 1(6), 2876-2890 (2006)]. These results indicated that the MCAP2, a combination of the LQA and GNNQQY sequences, has amyloidogenicity.

Figure R5. Amyloidogenicity assay of the MCAP2 fibrils. (a) and (b) Topological analysis of the MCAP2 fibrils by AFM. The height map image (left) of MCAP2 fibrils, and the topographic cross-sectional profile (right) taken through the white dashed line of the height map image. The image sizes are (a) 10 x 10 μm and (b) 5 x 5 μm, respectively. (c) and (d) TEM images of the MCAP2 fibrils. The black scale bar in TEM images is 2 μm in width. (e) CD spectra of the MCAP2 fibrils and monomers. The MCAP2 fibrils were fabricated by incubating 250 μg·mL⁻¹ of MCAP2 monomers at 37 °C with shaking at 1000 Hz for 5 days.

The AFM analysis of MCAP3 aggregates showed the absence of a fibril-like structure (Fig. R6a). The CD spectra of MCAP3 aggregates displayed a peak near 190 cm⁻¹, representing the random coil structure [(BBA)-Proteins and Proteomics, 1844(12), 2331-2337 (2014)] (Fig. R6b). These observations indicate that the MCAP3 aggregates do not possess fibrillar structure and have no amyloidogenicity compared to MCAP and MCAP2.

Figure R6. Amyloidogenicity assay of MCAP3 fibrils. (a) Topological analysis of the MCAP3 fibrils by AFM. The height map image (left) of MCAP3 fibrils, and the topographic cross-sectional profile (right) taken through the white dashed line of the height map image. (b) CD spectra of the MCAP3 fibrils and monomers. The MCAP3 aggregates were fabricated by incubating 250 μg·mL⁻¹ of MCAP2 monomers at 37 °C with shaking at 1000 Hz for 5 days.

We further conducted ThT analysis to confirm the proteolytic cleavage of M^{Pro} at the MCAP, the MCAP2, and the MCAP3 (Fig. R7). Each sample (100 μM, 80 μL) was incubated with 20 μL of M^{Pro} solution (1.48 μM) for 4h. The results showed a 42% reduction in the ThT fluorescence intensity for the MCAP fibrils and a 22% reduction for the MCAP2 fibrils by M^{Pro}, respectively. In contrast, the

MCAP3 aggregates exhibited no significant signal change when reacting with M^{pro}. These results suggested that among three, both variants of GNNQQY combined with LQS and LQA exhibit amyloidogenicity. In particular, the MCAP containing LQS shows the highest degradability to M^{pro}. We added this information in our revised manuscript (page 8, lines 8 to 19) and supporting information (Supplementary Fig. S5, S6, and S10).

Figure R7. ThT intensity changes of (a) the MCAP fibrils, (b) the MCAP2 fibrils, and (c) the MCAP3 aggregates when reacting with M^{pro}. The ThT fluorescence intensities were measured by excitation at 444 nm and emission at 510 nm in wavelength.

2. There are numerous amyloid peptides. What is the rationale to choose GNNQQY. The authors needs to include one more or more control amyloid to show the superior performance of the current one in the developed assay.

Our response: Thank you for your comment. As per your comment, we have designed various engineered sequences derived from amyloid-forming sequences such as islet amyloid polypeptide (IAPP, SNNFGAIL) [*Proceedings of the National Academy of Sciences*, 101(1), 87-92 (2004)] and amyloid-beta (KLVFFAE, GGIVIA) [*Protein Science*, 19(2), 327-348 (2010)]. The IAPP-derived engineered amyloid sequence is SNLQSNFGAIL (i.e., IAPP MCAP), and amyloid-beta-derived sequences are KLLQSVFFAE (i.e., A β MCAP1) and GGLQSVVIA (i.e., A β MCAP2) (Table. R2).

Name	Originated protein	Originated sequence	Engineered sequence
MCAP	Prion protein	GNNQQNY	LQGNLQSNQQNY
IAPP MCAP	IAPP protein	SNNFGAIL	SNLQSNFGAIL
Amyloid beta MCAP1	Amyloid beta protein	KLVFFAE	KLLQSVFFAE
Amyloid beta MCAP2	Amyloid beta protein	GGIVIA	GGLQSVVIA

Table. R2. Engineered amyloid sequences based on various amyloid-forming sequences and M^{pro}-cleavage sequences (LQS).

To confirm the amyloidogenicity of each engineered peptide, we conducted AFM, TEM imaging, and CD analysis (Fig. R8 to R10). The AFM and TEM images showed that fibrous aggregates were observed in each sample. The CD spectra of engineered amyloids have peaks near 220 cm^{-1} and 190 cm^{-1} , representing that they have a beta-sheet structure. These results indicated that even after inserting the M^{pro} cleavage sequence (LQS) into the amyloid-forming sequences (SNNFGAIL, KLVFFAE, and GGVVIA), they retain amyloidogenicity.

Figure R8. Amyloidogenicity assay of the IAPP MCAP fibrils (a) and (b) Topological analysis of IAPP fibrils by AFM. The height map image (left) of IAPP fibrils, and the topographic cross-sectional profile (right) taken through the white dashed line of the height map image. The images are (a) 10 x 10 μm and (b) 5 x 5 μm in size, respectively. (c) and (d) TEM images of the IAPP fibrils. The black scale bar in each TEM image is 2 μm in width. (e) CD spectra of the IAPP MCAP fibrils and monomers. The IAPP MCAP fibrils were fabricated by incubating $250\text{ }\mu\text{g}\cdot\text{mL}^{-1}$ of monomers at $37\text{ }^\circ\text{C}$ with shaking at 1000 Hz for 5 days.

Figure R9. Amyloidogenicity assay of the Aβ MCAP1 fibrils (a) and (b) Topological analysis of the Aβ MCAP1 fibrils by AFM. The height map image (left) of Aβ MCAP1 fibrils, and the topographic cross-sectional profile (right) taken through the white dashed line of the height map image. The images are (a) 10 x 10 μm and (b) 5 x 5 μm in size, respectively. (c) and (d) TEM images of the Aβ MCAP1

fibrils. The black scale bar in each TEM image is 2 μm in width. (e) CD spectra of the A β MCAP1 fibrils and monomers. The A β MCAP1 fibrils were fabricated by incubating 250 $\mu\text{g}\cdot\text{mL}^{-1}$ of monomers at 37 $^{\circ}\text{C}$ with shaking at 1000 Hz for 5 days.

Figure R10. Amyloidogenicity assay of the A β MCAP2 fibrils (a) and (b) Topological analysis of the A β MCAP2 fibrils by AFM. The height map image (left) of A β MCAP2 fibrils, and the topographic cross-sectional profile (right) taken through the white dashed line of the height map image. The images are (a) 10 x 10 μm and (b) 5 x 5 μm in size, respectively. (c) and (d) TEM images of the A β MCAP2 fibrils. The black scale bar in each TEM image is 2 μm in width. (e) CD spectra of the A β MCAP2 fibrils and monomers. The A β MCAP2 fibrils were fabricated by incubating 250 $\mu\text{g}\cdot\text{mL}^{-1}$ of monomers at 37 $^{\circ}\text{C}$ with shaking at 1000 Hz for 5 days.

To assess the reactivity between M^{Pro} and each amyloid sequence, a ThT assay was conducted (Fig. R11). Each sample at 100 μM was incubated with 1.48 μM of M^{Pro} solution for 4 h. The results showed that after the incubation with M^{Pro}, the ThT fluorescence intensity of the MCAP, the IAPP MCAP, the A β MCAP1, and the A β MCAP2 fibrils decreased by 42%, 26%, 17%, and 23%, respectively. From this result, we selected the MCAP as a substrate for our drug screening platform due to its superior sensitivity in assessing the proteolytic activity of M^{Pro}, compared to other engineered amyloid sequences. We added this information in our revised manuscript (page 8, line 20 to page 9, line 10) and supporting information (Supplementary Fig. S7 to S10)

Figure R11. ThT intensities of (a) the MCAP fibrils, (b) the IAPP MCAP fibrils (c) the A β MCAP1 fibrils, and (d) the A β MCAP2 fibrils before and after reacting with M^{Pro}. The ThT fluorescence intensities were measured by excitation at 444 nm and emission at 510 nm in wavelength.

3. The authors have demonstrated the amyloid fibril degrading ability of M^{pro}. The fibrils are known to assemble with strong hydrophobic and hydrogen bonding interaction results in embedding the core structure of peptides in the interior of the fibrils. How is the enzyme active site accessing the cleavage sequence in the core of peptide in fibrils structure? The enzyme active sites are very specific with precise special arrangements that allow them to fit the target polypeptide sequence. There is a need to address this by designing some experiments or computation.

Our response: Thank you for your insightful comment. To confirm that M^{pro} cleaves the core of the peptide in MCAP fibrils, we designed a ‘Repositioned MCAP’ wherein the M^{pro}-cleavage sequence is positioned at the end of the amyloid sequence, not in the core (Table R3). We assumed that if the M^{pro}-cleavage sequence is not present in the core of the peptide, the fibrils may not respond to M^{pro}. Validating this assumption allows us to demonstrate that the disassembly of MCAP fibrils is a result of the core sequence cleavage by M^{pro}.

Name	Originated protein	Originated sequence	Engineered sequence
MCAP	Prion protein	GNNQQNY	LQGNLQSNQQNY
Repositioned MCAP	Prion protein	GNNQQNY	LQGNNQQNYLQG

Table. R3. Engineered amyloid sequences comprising of prion protein-based amyloid-forming sequences and M^{pro}-cleavage sequence. At the Repositioned MCAP, the M^{pro}-cleavage sequence was placed at the end of the amyloid-forming sequence.

To validate the amyloidogenicity of Repositioned MCAP, we conducted AFM analysis, TEM imaging, and CD assay (Fig. R12). We observed fibrous aggregates of each sample in AFM analysis, and this was confirmed by TEM images. The CD spectra of the Repositioned MCAP fibrils showed peaks near 220 cm⁻¹ and 190 cm⁻¹, indicative of a beta-sheet structure. These results indicated that engineered peptides wherein the M^{pro} cleavage sequence (LQS) is attached to the end of amyloid-forming sequences have amyloidogenicity.

Figure R12. Amyloidogenicity assay of the Repositioned MCAP fibrils (a) and (b) Topological analysis of the Repositioned MCAP fibrils by AFM. The height map image (left) of Repositioned MCAP fibrils, and the topographic cross-sectional profile (right) taken through the white dashed line of

the height map image. The image sizes are (a) 10 x 10 μm and (b) 5 x 5 μm , respectively. (c) and (d) TEM images of the Repositioned MCAP fibrils. The black scale bar in TEM images is 2 μm in width. (e) CD spectra of the Repositioned MCAP fibrils and monomers. The Repositioned MCAP fibrils were fabricated by incubating 250 $\mu\text{g}\cdot\text{mL}^{-1}$ of monomers at 37 $^{\circ}\text{C}$ with shaking at 1000 Hz for 5 days.

We conducted a reactivity test between M^{Pro} and Repositioned MCAP fibrils through AFM analysis (Fig. R13). The results showed that their fibrillar structure remained regardless of M^{Pro} treatment, indicating that the Repositioned MCAP fibrils did not react with M^{Pro} (Fig. R13a and 13b). Also, the height of Repositioned MCAP fibrils treated with M^{Pro} revealed no significant changes compared to the non-treated (Fig. R13c). In addition, the CD spectra of the Repositioned MCAP fibrils with M^{Pro} exhibited no peak change in comparison with the Repositioned MCAP fibrils without M^{Pro} , indicating that their beta-sheet structure remained well in the presence of M^{Pro} . The ThT intensity of the Repositioned MCAP fibrils with M^{Pro} was reduced by 10% (Fig. R13e). These results demonstrated that the Repositioned MCAP fibrils have no M^{Pro} reactivity. The reason why this happens can be explained as follows: even though the M^{Pro} retains the cleavage activity at the end of Repositioned MCAP, the full sequence of GNNQQNY is still preserved after the M^{Pro} reaction, resulting in no degradation of the fibrillar structure. In the case of original MCAP, however, the M^{Pro} attacks the core region of original MCAP, and splits it into peptide fragments that are easily detached from the MCAP fibrils. This behavior may result in the continuous degradation of MCAP fibrils with time. These were confirmed by CD spectra in Fig. R14 that indicative beta-sheet peaks (220 cm^{-1} and 190 cm^{-1}) of MCAP fibrils decreased after M^{Pro} reaction, but Repositioned MCAP remained.

In this regard, we confirmed that M^{Pro} degrades only the MCAP fibrils which contain the M^{Pro} -cleavage sequence at the core region of the engineered sequence. We added and discussed this information in our revised manuscript (page 9, lines 11 to 21) and supporting information (Supplementary Fig. S11, and S12)

Figure R13. M^{PTO} reactivity test of the Repositioned MCAP fibrils (a) Topological analysis of the Repositioned MCAP fibrils by AFM. (b) Topological analysis of the M^{PTO}-treated Repositioned MCAP fibrils by AFM. (c) Height analysis of the Repositioned MCAP fibrils with/without M^{PTO} treatment. The height of each fibril was measured by NX10 software (Park Systems, South Korea). (d) CD spectra of the Repositioned MCAP fibrils with/without M^{PTO} treatment. (e) ThT intensities of the Repositioned MCAP fibrils and the M^{PTO} treated Repositioned MCAP fibrils.

Figure R14. CD spectra comparison between the MCAP fibrils and Repositioned MCAP fibrils. (a) CD spectra of the MCAP fibrils with/without M^{PTO} treatment. (b) CD spectra of the Repositioned MCAP fibrils with/without M^{PTO} treatment.

4. The authors describe MCAP acquire β -sheet on the AuNPs driving the nucleation. The β -conformation on AuNPs which needs to be substantiated by CD spectroscopy. Because in the local environment on the surface of charged AuNPs, behaviour of the peptide is uncertain and unclear.

Our response: We appreciate your valuable comment. Following your comment, we attempt to measure the CD spectrum of MCAP-AuNPs to prove the β -strand conformation of their surface. However, due to the relatively low concentration of amyloid proteins compared to AuNPs, the CD spectra of MCAP-AuNPs were not measured [*Analytical Chemistry*, 87(13), 6455-6459 (2015)]. Instead, we opted for FT-IR analysis, which allows us to analyze the conformation of the protein attached to the surface of AuNPs. Prior to the FT-IR analysis, we removed the free MCAP aggregates in the supernatant of the MCAP-AuNP solution by centrifugation and deposited MCAP-AuNPs onto a silicon wafer (Fig. R1a). We measured the FT-IR spectra of the MCAP-AuNPs, revealing that the FT-IR spectra show peaks near 1620 cm^{-1} and 1660 cm^{-1} (Fig. R15), indicative of the β -sheet structure [*The Journal of Physical Chemistry B*, 117(15), 4003-4013 (2013)]. Thus, we have confirmed that the MCAP aggregates attached to the MCAP-AuNP have the β -sheet conformation. We added this information in our revised manuscript (page 10, lines 19 to 21) and supporting information (Supplementary Fig. S15).

Figure R15. FT-IR spectra of the MCAP-AuNPs on silicon and the silicon wafer only. The MCAP-AuNPs was deposited on the surface of the silicon wafer, and then measured by FT-IR spectrometer. The scanning range spans $1500\text{-}1700\text{ cm}^{-1}$ in wavelength and the spectra was 8 nm in resolution.

5. The DLS experiments indicates the reduction of negative charged AuNPs. The studies show 98% coating of AuNPs with MCAP. Still the negative charge persists to a good extent. Both are contradicting. Is the coating percentage being overestimated?

Our response: We sincerely appreciate your comment. We estimated the percentage of MCAP coating on AuNPs based on TEM images (Fig. R3). Further validation of the MCAP coating on AuNPs was performed by assessing each zeta potential of bare AuNPs, MCAP-AuNPs, and MCAP fibrils (Fig. R16). The zeta potential of bare AuNPs, MCAP-AuNPs, and MCAP fibrils were -53.92 mV, -36.54 mV, and -30.96 mV, respectively. The results indicate that the MCAP-AuNPs are more similar in zeta potential with the MCAP fibrils as compared to the bare AuNPs, thereby supporting successful MCAP-coating on the surface of AuNPs. We added this information to our revised manuscript (page 11, lines 4 to 8).

Figure R16. Zeta potential values of the bare AuNP, the MCAP-AuNPs, and the MCAP fibrils. (N.S.: $p \geq 0.05$, ***: $p < 0.005$).

6. For stability study, freeze thaw was used. The temperature is another important parameter in a realistic manner. Are the MCAP-AuNPs stable at higher temperature or at AuNPs can be employed for Mpro activity detection. How many freeze thaw cycles can they sustain without losing activity.

Our response: We appreciate your insightful comment. To evaluate the temperature-dependent stability of MCAP-AuNPs, we conducted a stability test under various temperature conditions, including 40 °C, and 60 °C. At 40 °C, the UV-vis spectra of MCAP AuNPs were unchanged depending on time (Fig. R17a, and R17b), representing that the MCAP-AuNPs are stable at this temperature. In contrast, at the 60 °C condition, UV-vis spectra of MCAP-AuNPs exhibited a red-shift up on time, and the relative absorbance ratio (A_{650}/A_{525}) also increased over time. These results indicated that the MCAP-AuNPs were unstable at the 60 °C conditions due to thermal denaturation. Accordingly, there is no problem in the use of MCAP-AuNPs at the temperature range of 30~40 °C that corresponds to the physiological temperature range of M^{Pro}. This information has been included in our revised manuscript (page 11, lines 9 to 12) and supporting information (Supplementary Fig. S16)

Figure R17. Temperature-dependent stability test of the MCAP-AuNPs. (a) UV-vis spectra of the MCAP-AuNPs at 40 °C depending on time. (b) Relative absorbances (A_{650}/A_{525}) of the MCAP-AuNPs at 40 °C with time. (c) UV-vis spectra of the MCAP-AuNPs at 60 °C depending on time. (b) Relative absorbances (A_{650}/A_{525}) of the MCAP-AuNPs at 60 °C with time. (N.S: $p \geq 0.05$, *: $p < 0.05$, **: $p < 0.01$, ***: $p < 0.001$, ****: $p < 0.0001$).

We conducted repeated freeze-thaw tests for MCAP-AuNPs in Fig. R18. The results showed that the UV-vis spectra and relative absorbance (A_{650}/A_{525}) remained unchanged until the fourth cycle of the freeze-thaw. After the fifth cycle, the UV-vis spectra gradually exhibited a red shift, with a dramatic change observed in the eighth cycle. In addition, we performed the M^{pro} activity test by MCAP-AuNPs before and after freeze-thaw, and observed that the degree of reaction by M^{pro} remains constant even after freeze-thaw. These results indicate that the MCAP-AuNPs are suitable for cryopreservation maintaining a constant reactivity by M^{pro} after thawing for use. We added this information in our revised manuscript (page 11, lines 17 to 22) and supporting information (Supplementary Fig. S17).

Figure R18. Repeated freeze-thaw test of the MCAP-AuNPs. (a) UV-vis spectra of the MCAP-AuNPs depending on freeze-thaw cycle. (b) Relative absorbances (A_{650}/A_{525}) of the MCAP-AuNPs depending on freeze-thaw cycle. (c) The values of A_{650}/A_{525} of the MCAP-AuNPs before and after M^{pro} reaction depending on the concentration of M^{pro} . The freeze-thawed MCAP-AuNPs are marked in red and the normal MCAP-AuNPs are marked in blue. (N.S: $p \geq 0.05$, *: $p < 0.05$, **: $p < 0.01$).

7. The authors demonstrated the ability of MCAP-AuNPs to detect Mpro by aggregation of AuNPs. How to rule out the effect of Mpro to induce the aggregation by noncovalent interaction with MCAP-AuNPs. Suitable controls like BSA need to be used to check the ability of protein to induce MCAP-AuNPs aggregation. Vice versa, the effect of MCAP-AuNPs on the activity of Mpro needs to be ruled out by including known substrate for Mpro.

Our response: We appreciate your valuable comments. Accordingly, we conducted selectivity tests with various interfering molecules such as bovine serum albumin (BSA), immunoglobulin, glucose and human serum albumin (HSA) (Fig. R19 and Fig. R20). The UV-vis spectra of MCAP-AuNPs incubated with $0.1 \text{ mg} \cdot \text{mL}^{-1}$ and $1 \text{ mg} \cdot \text{mL}^{-1}$ of BSA were unchanged with time (Fig. R19a and R19b). Furthermore, the relative absorbance (A_{650}/A_{525}) of MCAP-AuNPs for each condition was unchanged even after 24h (Fig. R19c), representing that the MCAP-AuNPs were not aggregated. Thus, the MCAP-AuNPs have no reactivity for BSA, and this information was added to our revised manuscript (page 13, lines 10 to 14) and supporting information (Supplementary Fig. S19)

Figure R19. BSA stability test of the MCAP-AuNPs. (a) UV-vis spectra of the MCAP-AuNPs incubated with $0.1 \text{ mg}\cdot\text{mL}^{-1}$ of BSA depending on time. (b) UV-vis spectra of the MCAP-AuNPs incubated with $1 \text{ mg}\cdot\text{mL}^{-1}$ of BSA depending on time. (c) The values of A_{650}/A_{525} of the MCAP-AuNPs reaction with various concentrations (0, 0.1, and $1 \text{ mg}\cdot\text{mL}^{-1}$) of BSA after 24h. (N.S: $p \geq 0.05$).

Further, we performed a selectivity test of MCAP-AuNPs with several types of biomolecules abundantly existing in physiological conditions, including HSA, glucose, and immunoglobulin G (IgG) (Fig. R20). The reaction concentration of each molecule was chosen by considering its physiological range. Fig. R20a to c showed that the UV-vis spectra of MCAP-AuNPs treated with each molecule were unchanged with time. The MCAP-AuNPs reacted with each molecule for 24 h showed no significant change in the relative absorbance (A_{650}/A_{525}), compared to the MCAP-AuNPs with M^{pro} (Fig. R20d). These results indicate that the MCAP-AuNPs are not aggregated by other interfering molecules abundant in physiological conditions and selectively reacted with M^{pro} . We added this information in our revised manuscript (page 13, lines 10 to 14) and supporting information (Supplementary Fig. S19).

Figure R20. Stability test of the MCAP-AuNPs with albumin, glucose and IgG. UV-vis spectra of the MCAP-AuNPs reacted with (a) $30 \text{ mg}\cdot\text{mL}^{-1}$ of HSA (b) $1 \text{ mg}\cdot\text{mL}^{-1}$ of glucose, and (c) $10 \text{ mg}\cdot\text{mL}^{-1}$ of IgG depending on time. (d) Relative absorbances (A_{650}/A_{525}) of the MCAP-AuNPs treated with different types of biomolecules and M^{pro} , respectively, for 24 h. The concentrations of M^{pro} and denatured M^{pro} are 18.5 nM and the concentration of BSA is $1 \text{ mg}\cdot\text{mL}^{-1}$.

To assess the effect of substrate type on M^{pro} activity, fluorescence resonance energy transfer (FRET)-based measurement is typically employed. We measured the M^{pro} activity using the commercialized FRET substrate MCA-AVLQSGFR-Lys(Dnp)-Lys-NH₂ trifluoroacetate (Sigma Aldrich, USA) (Fig.

R21). We obtained the values of half-maximal effective concentration (EC_{50} , 211 nM) from the sigmoidal dose-response curve as a function of M^{pro} , which is approximately 50 times higher, compared to EC_{50} with our MCAP-AuNP platform (4.4 nM). Furthermore, we observed that the MCAP-AuNP-based system displayed a much lower variance for each data point, compared to the FRET-based system. These results suggest that our MCAP-AuNP-based system offers superior sensitivity and accuracy in measuring the M^{pro} activity compared to the commercialized FRET assay. It also drastically reduces the amount of a very expensive enzyme (M^{pro} , \$76,800/10 mg) that is required for screening. This information has been incorporated into our revised manuscript (page 13, lines 16 to 24 and page 14, line 1).

Figure R21. Concentration-dependent M^{pro} activity test by using the FRET-based assay and our MCAP-AuNP platform. The substrate concentration in FRET-based assay was 16 μ M, and the MCAP concentration in our MCAP-AuNP platform was 10 μ M.

8. In the drug screening assay, the possibility of drugs interacting with MCAP-AuNPs by non-covalent interactions preventing M^{pro} to degrade MCAP-AuNPs exists. Need to include appropriate control to rule out this possibility. How suitable this assay for HTS need to be discussed.

Our response: Thank you for your comment. We admit that there is a possibility that some drugs from a chemical library themselves interact with MCAP-AuNPs, not M^{pro} , resulting in false positive. To prevent any such false positive case, we added the following procedure in the screening.

Prior to an interaction between M^{pro} and MCAP-AuNP, M^{pro} was pretreated with drugs to inhibit their activity for 20 min. Subsequently, the inhibited M^{pro} interacted with MCAP-AuNPs to assess the efficacy of drugs by colorimetric responses of MCAP-AuNP solutions. This step may help to decrease the rates of false positive.

There are several conventional drug screening methods such as FRET and cell-based assay. However, they require the use of additional chemicals (e.g. fluorescent dyes), complicated sample preparation, long analytical time (~1 day: FERT, ~7 days: cell-based assay), and very high cost (because of the use of large amount of M^{pro}). For instance, 10 mg of M^{pro} is need to complete the screening of ~100 chemicals for FERT and ~10 chemicals for cell-based assay. Surprisingly, our MCAP-AuNP platform needs only 10 mg of M^{pro} to complete the screening of ~10,000 chemicals. Furthermore, the analytical time (just 1~2 hours) of our platform is much shorter than those of FRET and cell-based assay. The screening process is very simple by using colorimetric response and can be conducted only with 384-well plate, microplate reader and the MCAP-AuNPs. The advantages of our platform in the high-throughput screening (HTS) process are discussed in detail in our revised manuscript (page 20, lines 5-25; page 21, lines 1 to 4)."

REVIEWERS' COMMENTS

Reviewer #1 (Remarks to the Author):

The authors have properly revised manuscript according to the reviewers' comments. Thus, I recommend the publication of the manuscript in Nature Communications without further reievew.

Reviewer #2 (Remarks to the Author):

My major concerns with the manuscript have been addressed.

Reviewer #3 (Remarks to the Author):

The authors have carefully revised the manuscript by addressing all the reviewers' comments. The quality of the manuscript improved significantly and I recommend its publication.